# Inadequate foundational decoding skills constrain global literacy goals for pupils in low- and middle-income countries

Michael Crawford[1], Neha Raheel [1], Maria Korochkina [2] & Kathleen Rastle [2]✉

Learning to read is the most important outcome of primary education. However, despite substantial improvements in primary school enrolment, most students in low- and middle-income countries (LMICs) fail to learn to read by age 10. We report reading assessment data from over half a million pupils from 48 LMICs tested primarily in a language of instruction and show that these pupils are failing to acquire the most basic skills that contribute to reading comprehension. Pupils in LMICs across the first three instructional years are not acquiring the ability to decode printed words fluently and, in most cases, are failing to master the names and sounds associated with letters. Moreover, performance gaps against benchmarks widen with each instructional year. Literacy goals in LMICs will be reached only by ensuring focus on decoding skills in early-grade readers. Effective literacy instruction will require rigorous systematic phonics programmes and assessments suitable for LMIC contexts.

Learning to read is the most important outcome of a child's primary school education. If a child does not learn to read in primary school, then they will not be able to use reading to access the curriculum in secondary and tertiary education. High levels of literacy promote human capital accumulation, thus promoting income, employment, health benefits, and social participation for individuals, and poverty reduction and economic development for societies[1]. However, research indicates that approximately 57% of children in low- and middle-income countries (LMICs) are unable to read with a basic level of comprehension by age 10 (refs. 1,2). This staggering figure comes despite record investments in global education over the past 20 years. LMICs spend between 3.5% and 4.3% of gross domestic product on education, and education is a major target for international development assistance[3]. These investments have supported substantial rises in primary school enrolments to over 90% globally but have not brought meaningful increases in learning and literacy[4].

Research on low literacy achievement in LMICs has principally focused on systemic factors, such as teacher career progression structures, monitoring and accountability measures, school management capacity and classroom infrastructure[1,5,6]. Our approach instead considers the problem in relation to the science of reading. There is a large body of psychological research on how children learn to read and how they can best be taught[7]. This research has made major inroads into literacy policy and practice in high-income countries but has had limited impact in LMICs so far. One major insight from this research is that learning to read in an alphabetic writing system requires mastery of a range of lower-level reading subskills[7]. Most crucially, pupils must acquire the understanding that letters represent sounds, and they must learn to retrieve letter-to-sound mappings fluently to decode printed words. This decoding process allows pupils to use their spoken vocabulary to access the meanings of unfamiliar printed words[7], and it provides the basis for developing readers to get the reading practice vital for building proficiency through the later primary and secondary school years[8]. There is a strong consensus that the development of decoding skills requires high-quality, systematic phonics in the initial years of reading instruction[7].

[1]Global Education Practice, World Bank, Washington DC, USA. [2]Department of Psychology, Royal Holloway, University of London, Egham, UK. ✉e-mail: Kathy.Rastle@rhul.ac.uk

In high-income countries, early-reading curricula often have a strong focus on the development of decoding skills through systematic phonics[7], and these skills are often assessed and reported as part of the instructional programme using standard instruments such as the Dynamic Indicators of Basic Early Literacy Skills (DIBELS)[9] or through national phonics screening[10]. In fact, a majority of US states now have legislation in place requiring early screening for reading disabilities, with most making reference to the assessment of decoding skills (including 17 states that specifically mention use of DIBELS instruments[11]). In contrast, the Global Proficiency Framework[12] is the main framework agreed by all major international development agencies for stating reading progress targets in LMICs and it is focused largely on comprehension. Although this framework makes some reference to decoding, the word 'phonics' does not appear in its 146 pages, and it appears to suggest that new letter–sound correspondences may continue to be introduced any time from grade 1 to grade 9. Moreover, much of the data formally recorded in LMICs relates to schooling inputs and enrolments rather than learning outcomes[13], and there is a dearth of information reported on pupils' literacy, including whether they are acquiring basic decoding skills[2].

The absence of regular assessments of decoding for pupils in LMICs poses a challenge to understanding the poor reading outcomes in these countries. However, it is possible to evaluate the decoding skills of pupils in LMICs using the Early Grade Reading Assessment (EGRA)[14], a suite of reading tasks and assessment protocols modelled on DIBELS but designed for use in LMICs. The EGRA was created by the US Agency for International Development (USAID) and partner agencies to permit accurate assessment of student reading achievement. It has been used to measure the effectiveness of reading interventions, to guide instructional programmes and in academic research (see ref. 15 for a review). The EGRA differs from assessments specifically designed for cross-country comparisons in high-income countries, such as the Progress in International Reading Literacy Study and the Programme for International Student Assessment (which measure reading comprehension differences across countries at age 10 and 15, respectively), in that it is not overseen by a centralized assessment agency. However, the EGRA's strong theoretical basis, design standards and consistent procedures[14] allow for the construction of instruments that meet high psychometric standards for assessing pupils' foundational reading skills in a way that is comparable (if not psychometrically equivalent) across different countries and for different languages[15].

Our work brings together hundreds of EGRA surveys conducted in LMICs over a period of 15 years that have assessed over half a million pupils primarily in a language of instruction. This comprehensive aggregation of EGRA surveys provides a systematic organization of EGRA data in terms of reading subskills across the first three instructional years for a variety of languages and countries. Our aim is to assess whether children are acquiring the foundational decoding skills to become successful readers. We focus on four EGRA decoding tasks for which there are comparable DIBELS measures: letter name identification, letter sound identification, non-word reading and oral reading fluency (in passages). Scores indicate the number of letters, letter sounds, non-words (also known as pseudowords such as 'vib' or 'slint') and words read accurately in 1 minute. Our analyses sought to quantify decoding performance across the first three instructional years against benchmarks for the corresponding DIBELS tasks, seeking to assess whether pupils from LMICs are on a trajectory to becoming proficient readers. Our final analysis investigated the relationship between performance on the four decoding tasks and reading comprehension.

## Results

Our results showed strikingly poor performance on all measures of decoding and, critically, that performance tends to fall further below the benchmarks required for proficient reading with each instructional year. In the following, we describe performance on the four EGRA

decoding tasks before turning to statistical analyses investigating (1) pupil progress across instructional years, (2) deviation from benchmarks with each instructional year, and (3) the relationship between decoding performance and reading comprehension.

Figure 1 summarizes the findings of 230 EGRA surveys comprising 694 subsurveys undertaken in 48 countries, 96 languages and 22,656 schools, involving 526,862 pupils learning to read in alphabetic writing systems. The numbers of schools and pupils included in the sample should be considered estimates because in 12 of the 230 EGRA surveys (5%) the reported sample sizes are not disaggregated across different language or instructional year cohorts. Of the EGRA surveys, 75% were given in a local or national language, and 25% were given in 1 of the 4 main former colonial languages (English, French, Spanish or Portuguese). The languages and countries included in the analysis, together with the relevant EGRA survey numbers for each language and country, are provided in Supplementary Tables 1 and 2, respectively. Each of the 2,373 data points in Fig. 1 represents an average score across pupils included in an EGRA subsurvey for a particular reading subskill assessment. These data are compared against benchmarks for the corresponding English language DIBELS tasks assessing these reading subskills in the USA (ref. 9, pages 123–124 for benchmarks). The 'substantial-risk' (black) criterion reflects the DIBELS benchmark goal for each task averaged across the instructional year. Pupils scoring below this benchmark are deemed to require additional strategic support beyond the core curriculum to reach proficiency goals. The 'severe-risk' (red) criterion reflects the DIBELS at-risk cut score for each task averaged across the instructional year. Pupils scoring below this criterion are deemed to be at particularly high risk of reading failure and to require intensive intervention. Benchmarks are available only where appropriate: the DIBELS test for letter name identification is not given in the third instructional year (as almost all pupils would be expected to know their letter names at that point) and the DIBELS test for oral reading fluency is not given in the first instructional year (as few pupils would be expected to read fluently at that point).

It is immediately apparent that most of the average subskill scores fall below both benchmarks. By year 3, between 64% and 99% of the subskill scores fall below the substantial-risk benchmark, and between 55% and 96% of the subskill scores fall below the severe-risk benchmark (Table 1).

The data visualized in Fig. 1 include subskill scores for EGRAs from all languages in the database written with alphabetic writing systems and from both nationally representative and non-nationally representative samples. Results are similar when including only those EGRA surveys with a nationally representative sample (Supplementary Fig. 1 and Supplementary Table 3) and when including only subskill scores from EGRA surveys testing pupils in English (Supplementary Fig. 2 and Supplementary Table 4). These additional descriptive statistics suggest that the poor performance shown in Fig. 1 cannot be attributed to sampling lower-performing student groups within the population or to the inappropriate application of English benchmarks to a wider range of languages. Results are also similar when including only those subskill scores for EGRA surveys that included all four decoding measures (Supplementary Fig. 3 and Supplementary Table 5), suggesting that the pattern visualized in Fig. 1 is not due to an anomaly driven by the choice of measures in particular EGRA surveys.

### Pupil progress across instructional years

The results of a statistical analysis showed that subskill scores improved with each instructional year. Subskill scores in year 2 were generally superior to those in year 1, and subskill scores in year 3 were generally superior to those in year 2 (Table 2). This analysis was undertaken using a Bayesian alternative to a t-test[16] because, unlike traditional frequentist approaches providing only point estimates (typically, a measure of central tendency such as the mean), Bayesian methods can inform on the range of most likely (termed, credible) values for the effect of

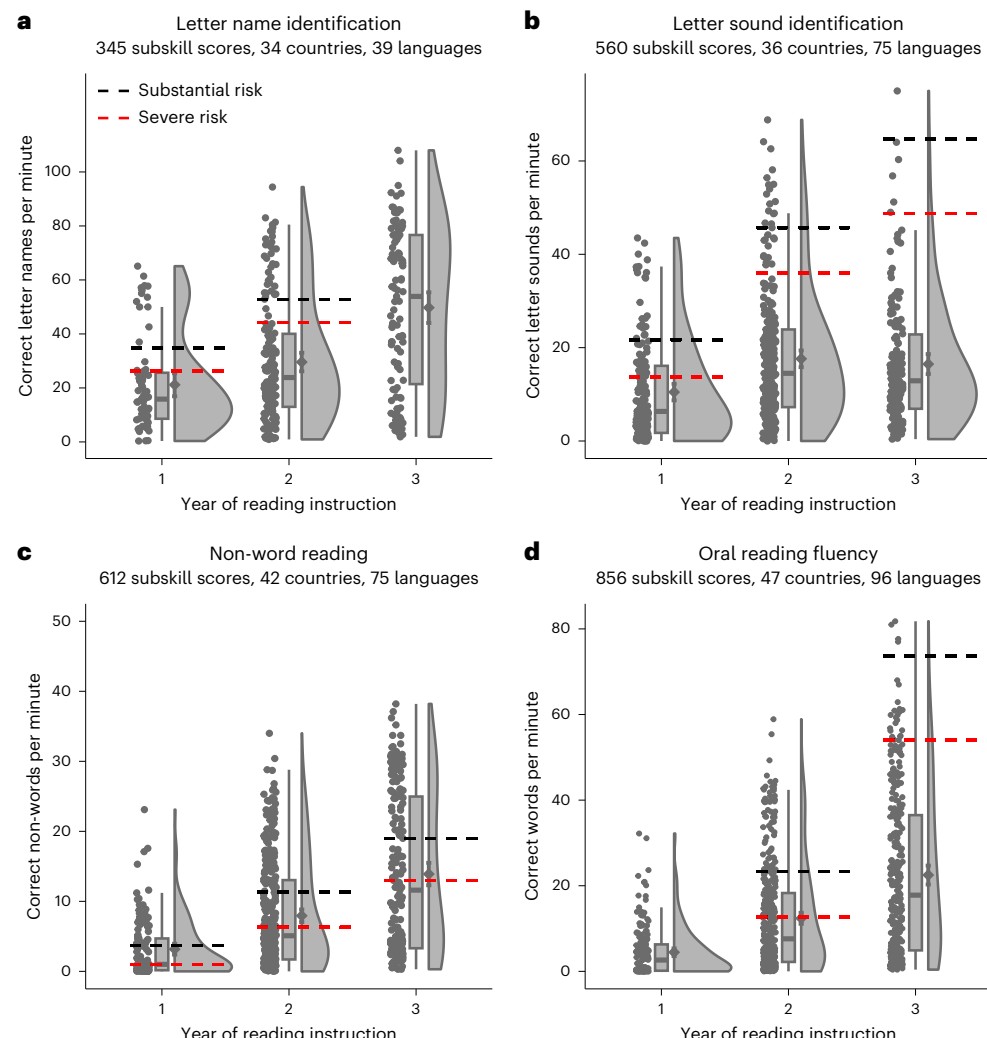

**Fig. 1 | Performance on four reading subskill measures for the first three instructional years. a–d,** The data from the four subskill measures are shown: letter name identification (**a**), letter sound identification (**b**), non-word reading (**c**) and oral reading fluency (**d**). For each plot, the number of subskill scores, countries and languages analysed is reported under the name of the task. Each plot shows the jittered raw data (the dots representing average subskill scores); the four quartiles of the ordered data (the box-and-whiskers plots) with the grey horizontal lines representing the medians (50%), the bounds of the boxes representing the lower and upper 25% quartiles, and the whiskers representing the expected variation of the data; and the estimated data distribution (the clouds) together with the means (the dots) and the 95% confidence intervals (the error bars). The DIBELS benchmarks (where available) for substantial and severe risk are represented using dashed lines (black and red, respectively) averaged across three time points (beginning, middle and end of year). The number of data points in each plot varies because not all EGRA surveys included all four decoding measures.

**Table 1 | Percentage of subskill scores falling below the substantial-risk and severe-risk benchmarks in each instructional year**

| Risk level and instructional year | Letter name identification | Letter sound identification | Non-word reading | Oral reading fluency |
|---|---|---|---|---|
| Substantial risk | | | | |
| Year 1 | 81% | 85% | 66% | No benchmark |
| Year 2 | 80% | 93% | 72% | 82% |
| Year 3 | No benchmark | 99% | 64% | 99% |
| Severe risk | | | | |
| Year 1 | 79% | 69% | 47% | No benchmark |
| Year 2 | 79% | 88% | 57% | 60% |
| Year 3 | No benchmark | 96% | 55% | 92% |

interest given the data (this range is referred to as the posterior distribution). This approach is thus both more robust and more informative than frequentist methods as it allows us to quantify uncertainty regarding the effects of interest[17,18].

The data in Table 2 show that, in almost all cases, there was improvement in average subskill scores across instructional years, as indicated by the fact that a difference of 0 was not typically among the 95% most credible differences of means. The only exception was for the comparison between year 2 and year 3 for letter sound identification, for which the 95% highest density credibility interval (95% HDI) did include 0, indicating no change in the average subskill score for that task.

The mean point estimates for most of the effect sizes in Table 2 are medium to large[19]. It is important to note that these effect sizes reflect changes in relative performance across the instructional years (the absolute scores remain very low; Fig. 1). However, the uncertainty around these mean point estimates is rather high (as evidenced by wide

**Table 2 | Differences in pupils' performance on each of the four decoding tasks across the three instructional years**

| Task | Contrast | Mean point estimate and 95% HDI of the posterior distribution | | |
| --- | --- | --- | --- | --- |
| | | Difference in means | Difference in s.d. | Effect size |
| Letter name identification (correct letter names per minute) | Year 2 versus year 1 | 8.49 [3.06, 13.80] | 4.58 [0.56, 8.57] | 0.47 [0.16, 0.78] |
| | Year 3 versus year 2 | 20.66 [14.10, 27.41] | 7.47 [2.78, 12.35] | 0.77 [0.51, 1.04] |
| Letter sound identification (correct letter sounds per minute) | Year 2 versus year 1 | 6.91 [4.69, 9.17] | 2.41 [0.59, 4.19] | 0.70 [0.45, 0.96] |
| | Year 3 versus year 2 | −1.30 [−3.70, 1.00] | −1.05 [−2.95, 0.85] | −0.13 [−0.37, 0.10] |
| Non-word reading (correct non-words per minute) | Year 2 versus year 1 | 4.39 [3.23, 5.54] | 3.50 [2.63, 4.39] | 0.76 [0.56, 0.95] |
| | Year 3 versus year 2 | 6.01 [4.19, 7.84] | 3.63 [2.35, 4.97] | 0.65 [0.45, 0.86] |
| Oral reading fluency (correct words per minute) | Year 2 versus year 1 | 6.37 [4.98, 7.77] | 5.52 [4.39, 6.68] | 0.82 [0.64, 0.99] |
| | Year 3 versus year 2 | 10.12 [7.65, 12.70] | 6.69 [4.88, 8.55] | 0.67 [0.50, 0.85] |

For each parameter, we report the mean point estimate (top row in the result columns) and the 95% HDI (bottom row) of the posterior distribution (that is, a range of credible values for the effect of interest). The effect size is Cohen's *d*[19] computed from the posterior distribution, with the mean representing the mean standardized distance between pupils' performance across the instructional years and the 95% HDI representing the uncertainty around this point estimate.

95% HDIs for the effect sizes), suggesting that there is high variability in performance across different education systems. The analysis of standard deviations in Table 2 suggests that this variability in performance tends to increase across instructional years, a pattern clearly visible in Fig. 1. These results suggest that the observed improvements in mean performance are most likely driven by improvements in a small proportion of education systems, while the majority remain at a low level of performance.

**Deviation from benchmarks across instructional years**
Despite the improvements documented in the previous analysis, a statistical comparison against the substantial-risk benchmarks[9] showed that performance gaps widen with each instructional year (a pattern clearly visible in Fig. 1). Using the same analysis as before[16], we compared the performance distribution expected under the substantial-risk benchmark (termed prior distribution) to the performance for each task and instructional year as observed in the data (Table 3). Priors were derived by simulating a normal distribution around the range of benchmark values for a particular task and instructional year (Methods). The effect size estimates for this comparison are large and negative, showing that, on average, performance is lower than the distribution expected under the substantial-risk benchmark in every case (that is, the 95% HDIs for the effect size estimates never include 0 or positive values). This analysis allows us to assert that there is 100% probability that the average predicted subskill scores (based on the posterior distribution) for each task and each instructional year will be credibly lower than the relevant average substantial-risk benchmarks. Critically, the absolute values for the effect size estimates increase across each instructional year for all tasks except for the non-word reading

task. These findings indicate that, for most of the subskills, as pupils progress through the first 3 years of reading instruction, the substantial-risk benchmark gets further out of reach. This result suggests that pupils in LMICs fall further from the trajectory required for proficient reading with each instructional year.

**Relationship between decoding and reading comprehension**
Our results showed a strong, positive correlation between all 4 decoding tasks and reading comprehension (all two-sided Pearson correlation coefficients (*r*) >0.60; Fig. 2). Data points were included only if scores were available in the EGRA database at the same time point for both reading comprehension and the relevant decoding task. Reading comprehension is assessed in the EGRA by asking literal and inferential questions about the oral reading fluency passage[14]. Reading comprehension subskill assessments in the EGRA database ranged from 3 to 20 items; however, to enhance the comparability of scores in Fig. 2, we include only those subskill assessments with between 4 and 6 items (this decision excludes no more than 8% of the eligible reading comprehension scores). Although the correlations depicted in Fig. 2 do not themselves establish a causal relationship, they are consistent with a wealth of evidence from psychological research on reading[7] that suggests that developing the ability to read for meaning requires acquisition of foundational decoding skills.

## Discussion

Every country in the world has committed to ensuring that all children are able to reach a minimum level of reading proficiency by 2030 (UN Sustainable Development Goals, Indicator 4.1.1) (ref. 20). Our results show that most children in LMICs are not even getting off the starting block. Only a small proportion of subskill scores in our EGRA database met minimum benchmarks for skills as basic as knowing the names and sounds associated with letters. Moreover, although there was evidence for some improvement in performance across each instructional year, this improvement was insufficient and subskill scores tended to drop further below benchmarks for successful reading with each instructional year. Our results indicate that pupils in LMICs are not developing the foundations of reading that will allow them to move onto a trajectory in which they become fluent, proficient readers able to acquire information from text.

The failure to master basic decoding skills in the initial years of reading instruction has clear consequences for reading comprehension. The ability to decode printed words fluently is vital given the extensive working memory and metacognitive demands associated with reading comprehension[7]. If a pupil struggles to decode each word, then they will also struggle to understand multi-word phrases or sentences and to undertake higher-level meaning computations that span many words (for example, inferences) and that may require integration with background knowledge[7]. Indeed, the Progress in International Reading Literacy Study assessment framework states that even the retrieval of explicitly stated information from text '… requires a fairly immediate or automatic understanding of the text' (ref. 21, page 21). The implication is that most of the samples in our EGRA database would not be able to engage at the most basic level with the main international assessment of reading comprehension. Our findings showing a strong, positive relationship between EGRA decoding subskills and reading comprehension assessment scores confirm these insights from psychological research on reading.

Our work is consistent with and provides further evidence for research on the dynamics of learning trajectories in LMICs. This work finds that the efficiency and effectiveness of additional education spending is at risk when student learning in the initial years of schooling is so low that it further endangers the ability to learn in subsequent school years[22]. Our data show that although some learning occurred in each instructional year, this learning was insufficient to stay on a trajectory of successful reading acquisition, making it increasingly difficult

**Table 3 | Performance on each task compared with that expected under the substantial-risk benchmark for each instructional year (prior distribution)**

| Task | Year | Prior distribution | Mean point estimate and the 95% HDI of the posterior distribution | | Substantial-risk benchmarks |
|---|---|---|---|---|---|
| | | | Predicted mean for the average subskill score | Effect size | |
| Letter name identification (correct letter names per minute) | 1 | Normal(35,5) | 22.43 [17.79, 26.93] | −0.74 [−1.15, −0.36] | 25, 37, 42 $M$=34.7 |
| | 2 | Normal(53,5) | 31.61 [28.23, 34.93] | −0.96 [−1.18, −0.76] | 42, 57, 59 $M$=52.7 |
| Letter sound identification (correct letter sounds per minute) | 1 | Normal(20,7) | 9.07 [6.85, 11.35] | −1.42 [−2.00, −0.92] | 9, 25, 31 $M$=21.7 |
| | 2 | Normal(45,7) | 15.80 [13.83, 17.76] | −2.71 [−3.32, −2.09] | 30, 52, 55 $M$=45.7 |
| | 3 | Normal(60,8) | 14.38 [12.24, 16.57] | −5.30 [−6.68, −3.97] | 50, 68, 76 $M$=64.7 |
| Non-word reading (correct non-words per minute) | 1 | Normal(3,1.5) | 0.48 [0.10, 1.16] | −5.25 [−9.44, −0.42] | 1, 3, 7 $M$=3.7 |
| | 2 | Normal(10,5) | 7.66 [6.68, 8.63] | −0.49 [−0.65, −0.34] | 5, 14, 15 $M$=11.3 |
| | 3 | Normal(18,0.7) | 16.27 [15.22, 17.37] | −0.24 [−0.34, −0.14] | 15, 20, 22 $M$=19 |
| Oral reading fluency (correct words per minute) | 2 | Normal(23,8) | 10.32 [8.56, 12.17] | −1.28 [−1.64, −0.91] | 10, 21, 39 $M$=23.3 |
| | 3 | Normal(73,13) | 22.50 [20.23, 24.69] | −2.71 [−2.99, −2.44] | 49, 78, 94 $M$=73.7 |

For each parameter, the prior was defined using the normal distribution (mean and s.d. are indicated in parentheses) and the results columns show the mean point estimate (top row) and the 95% HDI (bottom row) of the posterior distribution. The effect size is Cohen's $d$[19] computed from the posterior distribution, with the mean representing the mean standardized distance between performance expected under the substantial-risk benchmark and that achieved by the pupils, and the 95% HDI describing the uncertainty associated with this mean point estimate. The substantial-risk benchmark for each task and instructional year is reported in the rightmost column, with the top row showing the cut-off scores for the beginning, middle and end of each year, respectively, and the bottom row showing the average ($M$) score across these three time points.

for students to engage successfully with grade-level texts. The further into schooling a pupil moves without adequate reading skills, the more of the curriculum that pupil will be unable to access. Ultimately, the pupil may show no or low learning gains across the entire curriculum as a result of failing to acquire foundational reading skills.

Our findings therefore highlight the importance of measuring learning progress using absolute measures (such as words read aloud per minute) rather than effect sizes to judge progress. The learning gains that we observed from year to year generally had large effect sizes but were small in absolute terms when compared with the substantial-risk benchmarks. Large effect sizes from low baselines may give a false impression of meaningful progress in the growth of reading ability. To illustrate, oral reading fluency scores in our sample increased from 10 words per minute in the 2nd instructional year to 22 words per minute in the 3rd instructional year. This change yields a medium-to-large effect size of 0.67. However, the gain has little or no practical value because pupils of this age need to read at around 80 words per minute to exceed the substantial-risk benchmark. Substantial relative progress from low baselines still leaves pupils reading fewer than a third of the minimally acceptable words per minute required to stay on track for expected learning. These observations are consistent with findings showing that reading interventions in LMICs tend to have larger effect sizes than those in high-income countries; however, these interventions are typically associated with only modest gains in oral reading fluency and small numbers of pupils reaching proficiency due to low mean scores and a large proportion of non-readers at baseline[23].

One issue that requires discussion regards the appropriateness of US (English language) benchmarks. We chose DIBELS benchmarks because they have been informed by comparable reading assessments of millions of children (nearly a million children in 2019/2020 alone[24]) and have been shown repeatedly to have high technical adequacy[24]. No other empirically verified benchmarks relating these tasks to

standards for proficient reading currently exist. However, we acknowledge that this is a pragmatic decision and that it is important for the field to move towards benchmarks for individual languages or sets of languages that share meaningful features. One feature that has been described at length is spelling-to-sound transparency: learning to decode is easier in more transparent orthographies[25]. This probably has consequences for determining the instructional year in which a particular standard of decoding performance is expected. Similarly, languages can vary in whether they are transcribed using conjunctive or disjunctive orthographies (that is, whether meaningful units in a sentence are combined into single words or written separately) and this probably has consequences for determining expected oral reading fluency rates[26]. However, although the construction of language-specific benchmarks would permit a more refined evaluation of decoding performance in particular tasks, it is improbable that such benchmarks would undermine the general conclusions drawn from our analyses. We also note that English is one of the most difficult orthographies to learn due to its low degree of spelling-to-sound consistency[25], so if anything it provides a conservative benchmark for basic decoding tasks. Moreover, the analysis in which we restricted EGRAs to those pupils learning to read in English showed similar patterns of performance to the main analysis, suggesting that the pattern of performance depicted in Fig. 1 is unlikely to be due to the use of a benchmark that is inappropriate for the languages being tested.

Some questions might also be raised regarding the comparability of EGRA scores to US benchmarks given that children in LMICs frequently learn to read in multilingual environments and that around 37% of children in LMICs learn to read in a language other than the language spoken in their home[27]. Policy experts have argued that this poses serious additional challenges to pupils learning to read in LMICs[27], although we caution against framing this issue as one that arises only in LMICs (for example, over 20% of pupils in Texas were learning English

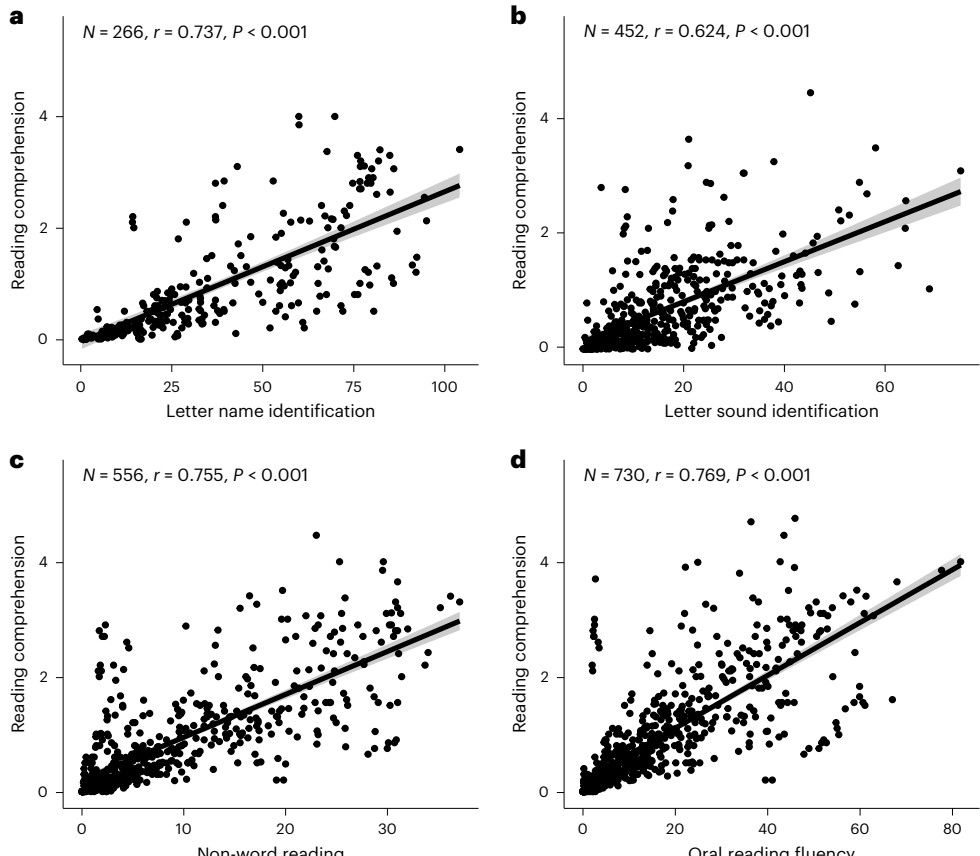

**Fig. 2 | Relationship between the performance on the four decoding tasks and on the reading comprehension task. a–d**, Scatter plots of the data from the reading comprehension task (*y* axis) and from one of the four decoding tasks (*x* axis): letter name identification (number of letters named accurately in one minute) (**a**), letter sound identification (number of letter sounds produced accurately in one minute) (**b**), non-word reading (number of non-words read aloud accurately in one minute) (**c**) and oral reading fluency (number of words read aloud accurately in one minute) (**d**). Reading comprehension scores were included only where the maximum possible number of items was between four and six. Two-sided Pearson *r* values and the associated *P* values are shown in the top-left corner of each panel. The grey shading around the linear regression lines represents the 95% confidence interval.

as an additional language in 2020[28]). Indeed, psychological research on reading suggests that learning to read requires both decoding and spoken language[7]. Even if a child has excellent decoding skills, if the spoken word is not in their vocabulary, they will not be able to compute meaning. However, although this language-of-instruction issue almost certainly impacts on comprehension, we are less convinced that it would have an impact on simple decoding tasks, such as learning the names and sounds associated with letters. We cannot evaluate the spoken language abilities of the samples depicted in Fig. 1 because, although the EGRA database contains listening comprehension scores, there is no equivalent DIBELS task or benchmark to measure these scores against. However, we investigated the relationship between listening comprehension and the reading measures reported in the 'Results' (note that listening comprehension is administered in an analogous manner to reading comprehension, that is, by asking literal and inferential questions about a spoken passage[14]). These analyses showed that although there is a significant relationship between reading comprehension and listening comprehension in these samples, it is far weaker (*r* = 0.26; Supplementary Fig. 4) than the relationship between reading comprehension and decoding subskills (*r* > 0.60 for each of the 4 subskills; Fig. 2). The relationship between performance on listening comprehension and decoding subskill tasks shows a similar pattern: it is statistically significant for 3 of the tasks (all but letter name identification) but weak in all cases (all *r* < 0.18; Supplementary Fig. 5). Thus, although research indicates that spoken language ability is an important foundation of decoding and reading comprehension[7], and

although this relationship is observed in the samples described in the EGRA database, deficiencies in spoken language are probably not a primary driver for the poor reading outcomes documented in this article.

One final point is that our conclusions are necessarily limited by the nature of the data itself: our analyses are based on the data that are available. We have already discussed the limitations of the EGRA, in that its construction and administration are not overseen by a central agency and thus deviations in protocols across surveys can arise. Likewise, the EGRA surveys and subsurveys used in our analyses differ in their sampling methodologies (with only a proportion being nationally representative) and they aggregate across different languages, instructional contexts and socio-economic circumstances. Finally, we only have access to averaged data for populations from each subsurvey; hence, although we can make inferences about performance at the systems level, we cannot make inferences about the reading proficiency of individuals. We sought to overcome these limitations by assembling a large quantity of data, scrutinizing the protocols of underlying data rigorously, undertaking control analyses where necessary, and adopting the most robust statistical approach possible given the nature of the data.

Overall, our findings show a major failure of public policy. School systems globally, including those in LMICs, seek to form capable readers as a cornerstone of the production of human capital. Psychological research strongly suggests that decoding is the foundation for reading comprehension and must be mastered in the initial years of reading instruction[7]. However, this body of research is not being translated effectively into the major instruments for stating and assessing

progress targets in LMICs, such as the Global Proficiency Framework, which is focused primarily on comprehension[12]. The standards for fluent reading of connected text, and for understanding and interpreting texts outlined in the Global Proficiency Framework, are not achievable without antecedent decoding skills. Our data show that instructional efforts in LMICs are failing to equip pupils with the most basic decoding skills necessary for individual word reading, thus undermining any instructional time spent on higher-level text comprehension.

The policy responses to our findings must shift efforts to improve reading instruction to its first critical point of failure so that children become fluent decoders by the third instructional year. The science of reading provides a clear pathway to addressing this challenge, namely, through development and deployment of rigorous systematic phonics programmes that provide explicit instruction on how to translate letters to sounds[7]. These instructional programmes should be accompanied by regular assessments of pupils' decoding ability. Such assessments are used increasingly in high-income countries to identify pupils in need of further support and to monitor the effectiveness of the instructional programme. The reading research community should support this ambition by moving to a global perspective that embraces the special challenges faced in LMICs. Research on reading acquisition and effective instruction in other widely used languages and writing systems, in multilingual settings and in low-resource contexts is urgently needed.

## Methods

### Database development
Existing EGRA surveys were acquired from 254 publicly available reports from the following sources: USAID's data repository[29], RTI International's Structured Pedagogy Literacy Review[30], RTI International's Learning at Scale Report[31], USAID's Early Grade Reading Barometer[32], and data repositories associated with Room to Read[33] and Save the Children[34]. No special permissions were needed to access these EGRA data. Ethical protocols for collecting EGRA data are described in the EGRA Toolkit[14] and many of the reports provide additional detail. Ethical review is not relevant to our analyses given that the data we used are (1) publicly available and (2) averaged across study populations (that is, no data from individual human participants were handled). The reference and link to each EGRA survey is available in the database, and the 254 reports are also archived in the Open Science Framework (OSF) site for this project (see 'Data availability'). This work was not preregistered.

The complete database comprises 347 EGRA surveys conducted between 2007 and 2021. The EGRA surveys in this database comprise data across the first 3 instructional years for 726,680 pupils in 32,800 schools in 62 LMICs in 115 languages tested primarily in a language of instruction. The complete dataset includes EGRA surveys using alphabetic ($N = 238$), abjad ($N = 62$) and alphasyllabic ($N = 57$) writing systems. However, the results reported in this article reflect surveys in alphabetic writing systems only. This restriction is necessary to more fairly compare performance against US (English language) benchmarks for equivalent tasks. The languages and countries included in our analyses are available in Supplementary Tables 1 and 2, respectively.

For the purposes of this database, an EGRA survey is defined as an assessment undertaken in a particular country at a particular time. EGRA surveys frequently include a number of subsurveys that may refer to different languages, instructional years, geographic regions, intervention groups or cohorts. The 347 EGRA surveys in this database include 1,015 subsurveys, each of which reports an average of 7 subskill scores, yielding a total of 7,345 subskill scores in the database. These subskill scores are the average for the population being tested in an EGRA survey or subsurvey; scores for individual pupils are not typically available in EGRA reports.

The dataset specifies whether the sampling strategy used in each EGRA survey allows classification as nationally representative ($N = 67$) or not ($N = 281$). The nationally representative EGRA surveys comprise

data from 32 LMICs in 49 languages, representing reading proficiency in 9,996 schools and in 202,335 pupils across the first 3 instructional years.

The results reported reflect four tasks associated with foundational decoding skills and their relation to reading comprehension. However, to facilitate wider secondary analyses, the database also includes subskill scores relating to listening comprehension, oral vocabulary, initial sound identification, phoneme segmentation, syllable identification, familiar word reading, orientation to print, dictation and maze (cloze). The number of subskill scores available for these additional tasks is available in Supplementary Table 6.

The EGRA Toolkit[14] provides detailed guidance on the construction and administration of each subskill assessment and describes protocols for ensuring psychometric robustness including piloting, conducting enumerator training, and assessing internal consistency and inter-rater reliability. The reports underpinning the EGRA database frequently provide this psychometric information. However, there can be variation in the rigour with which individual EGRA surveys are administered in LMICs. Thus, we scrutinized the data carefully to identify cases in which there were deviations from standard practice in administration. This scrutiny involved comparing the instrument or the description of the instrument used to generate each subskill score with the standard protocols for that task described in the EGRA Toolkit[14] to ascertain whether there were any substantive deviations. This process resulted in the removal of four subskill scores from each of the letter names and oral reading fluency datasets because the measures did not reflect a 'per minute' value. For future users of the EGRA database, we have also flagged 366 subskill scores for tasks not analysed in this study for which there were deviations in administration from standard protocols. These deviations in the administration of EGRA tasks are documented fully on the OSF site for this project.

The benchmarks used for comparison with the EGRA subskill scores were taken from the *DIBELS 8th Edition*[9]. The DIBELS benchmarks were produced and validated through research that established measures of reliability and of concurrent and predictive validity for all subtests at all grades. Predictive validity is established by frequently retesting students and comparing their scores during the year with end-of-grade scores on the Iowa Assessment of Total Reading and other tests of reading ability (see ref. 35 for further detail).

### Analysis approach
The OSF site for this project contains code for all analyses reported in this article (see 'Code availability'). The statistical analyses comparing performance across instructional years and those comparing performance to the substantial-risk benchmark were conducted through a Bayesian alternative to a $t$-test[16,36] (as implemented in the R package BEST v.0.5.4). The logic behind Bayesian inference is straightforward: we first specify plausible values for the effect of interest based on our prior knowledge (referred to as prior distribution) and, after having seen the data, assess by how much we need to change our beliefs such that they are consistent with those data (posterior distribution). Unlike the more traditional null hypothesis significance testing, when certainty in the estimates is high, the Bayesian framework allows us to accept the null value (for example, hypothesis of no difference between the two groups), rather than only to reject it, while it also provides complete distributional information about the samples and their differences, which is not possible under the frequentist approach[17,18,36]. Specifically, Bayesian methods allow estimating distributions of credible values for group means and standard deviations and their differences, and credible values for the effect size. Another advantage of computing Bayesian probability distributions is that they can be used to ascertain the probability that the true average score lies below (or above) a comparison value of interest[16,36]. We report such probabilities in the second analysis using the average substantial-risk benchmark as a comparison value.

In the R package BEST, sampling is done through a Bayesian Markov chain Monte Carlo (MCMC) process as implemented in the

JAGS software[37]. Data are described with the *t* distribution, which has fatter tails than the normal distribution and can thus better accommodate outliers. In analysis 1, comparing performance across instructional years, we used the set of priors suggested in ref. 16. For the mean parameters $\mu_1$ and $\mu_2$, a broad normal distribution was used: $\mu$ ~ Normal (mean(*y*),1,000 × s.d.(*y*)). Here, the mean is set to the mean of the pooled data *y*, and the s.d. is set to 1,000 times the s.d. of the pooled data. This setting ensures that the prior is scaled appropriately in relation to the scale of the data and that its influence on the estimates of the parameters is minimal such that, during parameter estimation, even a modest amount of data can overwhelm the prior assumptions. Similarly, the prior on the s.d. is also assumed to be non-committal, defined as a uniform distribution ranging from 0.001 of the s.d. of the pooled data to 1,000 times the s.d. of the pooled data: $\sigma$ ~ Uniform(s.d .(*y*)/1,000,s.d.(*y*) × 1,000). Finally, the normality of the data within the groups is expressed with the *v* parameter, which governs the relative height of the tails of the *t* distribution (when *v* is small, the *t* distribution has heavy tails and, when *v* is large, the *t* distribution is nearly normal) and thus permits description of data with and without outliers. This parameter has a prior that is distributed exponentially, $p(v|\lambda) = (1/\lambda)\exp[-(v-1)/\lambda]$, where $v \geq 1$ and $\lambda = 29$. The $\lambda$ parameter is set to 29 to balance nearly normal distributions ($v > 30$) with heavy-tailed distributions ($v < 30$); this prior distribution has been shown to perform better than other distributions (for example, uniform, shifted gamma, shifted or folded *t* distributions; see refs. 16,36 for a more detailed discussion).

In analysis 2, comparing performance to substantial-risk benchmarks, the priors were defined based on the performance expected for a given task and instructional year within the DIBELS framework for the pupils to not fall into the substantial-risk category (ref. 9, see pages 123–124 for benchmarks). For instance, for letter name identification in year 1, the DIBELS benchmark for substantial risk is 25 at the beginning of the year, 37 in the middle of the year and 42 at the end of the year, with 34.7 being the average benchmark score for that year. We therefore assumed a normally distributed prior with a mean of 35 and an s.d. of 5, reflecting a prior belief that if pupils are assessed at different times throughout this instructional year, 95% of the scores on this task should fall roughly between 25 and 45 (see Table 3 for other priors).

The descriptive model of the data used in the analyses is denoted mathematically as follows:

$$p(\mu_1, \sigma_1, \mu_2, \sigma_2, v|D) = p(D|\mu_1, \sigma_1, \mu_2, \sigma_2, v) \times p(\mu_1, \sigma_1, \mu_2, \sigma_2, v)/p(D)$$

In this equation, the posterior distributions on the parameters $\mu_1$, $\sigma_1$, $\mu_2$, $\sigma_2$ and *v* (expression to the left of the equals sign) are derived by multiplying the likelihood of this combination of parameters (first expression to the right of the equals sign) by the prior credibility of this combination (second expression to the right of the equals sign) divided by the evidence $p(D)$. The likelihood is defined as the multiplicative product of the data values of the probability density function of the *t* distribution, and the prior is defined as the product of distributions on $\mu_1$, $\sigma_1$, $\mu_2$, $\sigma_2$ and *v*. The value of the evidence constant is impossible to compute analytically as it requires integrating the product of the likelihood and prior over the entire parameter space. In BEST and other Bayesian approaches, this problem is solved by using Markov chain Monte Carlo sampling, which permits approximation of the posterior distribution by generating a large sample from it without explicitly computing the integral. The choice of the model structure was motivated by the features of the EGRA database, namely, that it contains observational data where, for each individual subskill assessment, each data point represents a score averaged across all pupils included in a given EGRA subsurvey (rather than a score achieved by each individual pupil). The sampling methods also differ across the subsurveys in the EGRA database; for example, different schools and/or students may have been assessed at different times within an instructional year, and subsurveys within the EGRA surveys used different languages,

geographic regions and cohorts. These attributes of the dataset make it difficult to define a model with a (more) complex hierarchical structure that would make appropriate assumptions regarding possible contributing variables and therefore a simpler model like the one used here is more suitable.

For both analyses 1 and 2, for each parameter of interest (mean, s.d. and effect size), we report the estimate of the posterior mean point estimate together with its 95% HDI, which denotes the range within which we can be 95% certain that the most credible values of the parameter fall. In analysis 1, the estimate of the effect size was computed for each credible combination of means and standard deviations from the posterior using the formula $(\mu_1 - \mu_2)/\sqrt{[\sigma_1^2(N_1-1) + \sigma_2^2(N_2-1)]/(N_1+N_2-2)}$, where $\mu_1$, $\sigma_1$, and $N_1$ and $\mu_2$, $\sigma_2$, and $N_2$ denote the mean, s.d. and the sample size of groups 1 and 2, respectively. Note that, contrary to the method adopted in ref. 16, we followed a more standard approach to computing the effect size, in which the estimate is weighted by the sample sizes of the two groups being compared[38]. In analysis 2, the following formula was applied: $(\mu - AB)/\sigma$, where $\mu$ and $\sigma$ denote the mean and the s.d. of the sample, and *AB* refers to the average benchmark for the relevant task and instructional year. These effect size estimates are thus equivalent to Cohen's $d$[19] for two-sample and one-sample *t*-tests but their added value is that, by using the entire posterior distribution rather than point estimates from the raw data, they provide information not only about the mean effect size but also about the uncertainty associated with it (the 95% HDI).

One reviewer suggested that the data may have been generated by a skewed, rather than a symmetric, process. To explore this possibility, we computed the Pearson's skewness coefficient for all variables used in the analysis (Supplementary Table 7). For all variables but one, the skewness coefficient was within the acceptable range for a symmetric distribution (−2 to +2) and, for four variables, the absolute value of the skewness coefficient was below 1, which is typically considered excellent[39,40]. Nonetheless, we remodelled the data using a skewed *t* distribution[41] for the likelihood function. The estimates produced by this analysis differ only slightly from those reported in the 'Results', confirming that the results remain unchanged even when the skew is accounted for. This additional analysis is reported in the Supplementary Appendix, and the code is available on OSF (https://osf.io/6s23f/).

### Reporting summary
Further information on research design is available in the Nature Portfolio Reporting Summary linked to this article.

## Data availability
The EGRA database and supporting information is available at https://osf.io/6s23f.

## Code availability
The code for all analyses is available at https://osf.io/6s23f/.

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

## Acknowledgements

This work was supported by a research grant from the Economic and Social Research Council (ES/W002310/1) awarded to K.R. The funder had no role in study design, data collection and analysis, decision to publish or preparation of the manuscript.

## Author contributions

M.C. conceptualized the study, was responsible for methodology, reviewed and edited the article and provided supervision. N.R. was responsible for methodology, investigation, and data curation, and reviewed and edited the article. M.K. was responsible for formal analysis, visualization, and data curation, and wrote the original draft. K.R. conceptualized the study, was responsible for methodology, wrote the original draft and provided supervision.

## Competing interests

The authors declare no competing interests.

## Additional information

**Correspondence and requests for materials** should be addressed to Kathleen Rastle.

**Peer review information** *Nature Human Behaviour* thanks Pamela Snow and the other, anonymous, reviewer(s) for their

contribution to the peer review of this work. Peer reviewer reports are available.

# Reporting Summary

## Statistics

For all statistical analyses, confirm that the following items are present in the figure legend, table legend, main text, or Methods section.

| n/a | Confirmed | |
|---|---|---|
| ☐ | ☒ | The exact sample size (*n*) for each experimental group/condition, given as a discrete number and unit of measurement |
| ☐ | ☒ | A statement on whether measurements were taken from distinct samples or whether the same sample was measured repeatedly |
| ☐ | ☒ | The statistical test(s) used AND whether they are one- or two-sided<br>*Only common tests should be described solely by name; describe more complex techniques in the Methods section.* |
| ☒ | ☐ | A description of all covariates tested |
| ☐ | ☒ | A description of any assumptions or corrections, such as tests of normality and adjustment for multiple comparisons |
| ☐ | ☒ | A full description of the statistical parameters including central tendency (e.g. means) or other basic estimates (e.g. regression coefficient) AND variation (e.g. standard deviation) or associated estimates of uncertainty (e.g. confidence intervals) |
| ☒ | ☐ | For null hypothesis testing, the test statistic (e.g. *F*, *t*, *r*) with confidence intervals, effect sizes, degrees of freedom and *P* value noted<br>*Give P values as exact values whenever suitable.* |
| ☐ | ☒ | For Bayesian analysis, information on the choice of priors and Markov chain Monte Carlo settings |
| ☒ | ☐ | For hierarchical and complex designs, identification of the appropriate level for tests and full reporting of outcomes |
| ☐ | ☒ | Estimates of effect sizes (e.g. Cohen's *d*, Pearson's *r*), indicating how they were calculated |

*Our web collection on statistics for biologists contains articles on many of the points above.*

## Software and code

Policy information about availability of computer code

**Data collection**

No software was used in data collection.

**Data analysis**

Software: All analyses were performed in R, version 4.2.1, platform x86_64-w64-mingw32/x64 (64-bit), running under Windows 10 x64 (build 19045)

Attached packages: kableExtra_1.4.0 , BEST_0.5.4, HDInterval_0.2.4, ggpubr_0.6.0, cowplot_1.1.3, RColorBrewer_1.1-3, reshape_0.8.9, scales_1.3.0, MASS_7.3-60, lubridate_1.9.3, forcats_1.0.0, stringr_1.5.1, dplyr_1.1.4, purrr_1.0.2, readr_2.1.5, tidyr_1.3.1, tibble_3.2.1, ggplot2_3.5.1, tidyverse_2.0.0, Rmisc_1.5.1, plyr_1.8.9, lattice_0.20-45, openxlsx_4.2.5.2, moments_0.14.1, bayesplot_1.11.1, rstan_2.32.6, StanHeaders_2.32.8

Packages loaded via a namespace (and not attached): Rcpp_1.0.12, svglite_2.1.3, digest_0.6.33, utf8_1.2.4, R6_2.5.1, backports_1.5.0, coda_0.19-4.1, evaluate_0.23, pillar_1.9.0, rlang_1.1.1, rstudioapi_0.16.0, car_3.1-2, rjags_4-15, rmarkdown_2.27, webshot_0.5.5, munsell_0.5.1, broom_1.0.6, compiler_4.2.1, xfun_0.40, systemfonts_1.1.0, pkgconfig_2.0.3, htmltools_0.5.8, tidyselect_1.2.1, gridExtra_2.3, viridisLite_0.4.2, fansi_1.0.6, tzdb_0.4.0, withr_3.0.0, ggdist_3.3.2, grid_4.2.1, distributional_0.4.0, gtable_0.3.5, lifecycle_1.0.4, magrittr_2.0.3, zip_2.3.1, cli_3.6.1, stringi_1.8.4, carData_3.0-5, ggsignif_0.6.4, xml2_1.3.6, generics_0.1.3, vctrs_0.6.5, tools_4.2.1, glue_1.7.0, hms_1.1.3, parallel_4.2.1, abind_1.4-5, fastmap_1.1.1, yaml_2.3.8, timechange_0.3.0, colorspace_2.1-0, rstatix_0.7.2, knitr_1.44 , jsonlite_1.8.8, QuickJSR_1.1.3,  xfun_0.40,  R6_2.5.1, pkgbuild_1.4.4, loo_2.7.0, inline_0.3.19, stats4_4.2.1, matrixStats_1.3.0

Our article refers to the Jags software and the Stan programming language. Both were accessed though R using the packages rstan and BEST (listed above) which integrate Jags and Stan with R, without the need to use either software directly.

The code for all analyses is available on https://osf.io/6s23f/.

For manuscripts utilizing custom algorithms or software that are central to the research but not yet described in published literature, software must be made available to editors and reviewers. We strongly encourage code deposition in a community repository (e.g. GitHub). See the Nature Portfolio guidelines for submitting code & software for further information.

## Data

Policy information about availability of data

All manuscripts must include a data availability statement. This statement should provide the following information, where applicable:
- Accession codes, unique identifiers, or web links for publicly available datasets
- A description of any restrictions on data availability
- For clinical datasets or third party data, please ensure that the statement adheres to our policy

The EGRA database and supporting information is available on https://osf.io/6s23f/.

## Research involving human participants, their data, or biological material

Policy information about studies with human participants or human data. See also policy information about sex, gender (identity/presentation), and sexual orientation and race, ethnicity and racism.

| | |
|---|---|
| Reporting on sex and gender | The data comprise publicly-available records of the reading skills of whole populations of pupils. No information about the sex or gender of individual pupils in these samples is available. |
| Reporting on race, ethnicity, or other socially relevant groupings | The population-level data on reading skills that we report is not broken down by race, ethnicity or any other socially relevant groupings. |
| Population characteristics | Participants were pupils in low- and middle-income countries in the first three years of reading instruction.  The data reported are an aggregation of population-level reading skills; for example, the reading skills of the whole population being tested in a particular set of schools in a particular instructional year. |
| Recruitment | Publicly available data were sourced from between 2007 and 2021 regarding the reading skills of pupils in low- and middle-income countries. These came from 254 separate reports from international development agencies and associated organisations.  The reports containing the underlying data are openly available in the OSF site for this project (https://osf.io/6s23f/). |
| Ethics oversight | This study involved aggregation of publicly-available data on the reading skills of whole populations of pupils between the periods of 2007 and 2021. The researchers do not have access to any information about individual pupils in these population-level datasets, and there are no data reported concerning individual pupils in this article. Therefore, ethics approval is not required. |

Note that full information on the approval of the study protocol must also be provided in the manuscript.

# Field-specific reporting

Please select the one below that is the best fit for your research. If you are not sure, read the appropriate sections before making your selection.

☐ Life sciences      ☒ Behavioural & social sciences      ☐ Ecological, evolutionary & environmental sciences

For a reference copy of the document with all sections, see nature.com/documents/nr-reporting-summary-flat.pdf

# Life sciences study design

All studies must disclose on these points even when the disclosure is negative.

| | |
|---|---|
| Sample size | Describe how sample size was determined, detailing any statistical methods used to predetermine sample size OR if no sample-size calculation was performed, describe how sample sizes were chosen and provide a rationale for why these sample sizes are sufficient. |
| Data exclusions | Describe any data exclusions. If no data were excluded from the analyses, state so OR if data were excluded, describe the exclusions and the rationale behind them, indicating whether exclusion criteria were pre-established. |
| Replication | Describe the measures taken to verify the reproducibility of the experimental findings. If all attempts at replication were successful, confirm this OR if there are any findings that were not replicated or cannot be reproduced, note this and describe why. |
| Randomization | Describe how samples/organisms/participants were allocated into experimental groups. If allocation was not random, describe how covariates were controlled OR if this is not relevant to your study, explain why. |
| Blinding | Describe whether the investigators were blinded to group allocation during data collection and/or analysis. If blinding was not possible, describe why OR explain why blinding was not relevant to your study. |

# Behavioural & social sciences study design

All studies must disclose on these points even when the disclosure is negative.

| | |
|---|---|
| Study description | This study aggregates publicly available data from between 2007 and 2021 on the reading skills of pupils in low- and middle-income countries and analyses these data against benchmarks for successful reading acquisition. The data are quantitative. |
| Research sample | The sample includes over half a million pupils learning to read in an alphabetic writing system across the first three years of reading instruction in low- and middle-income countries. This sample was derived by bringing together 230 existing EGRA (Early Grade Reading Assessment) surveys comprising 694 sub-surveys undertaken in 48 countries between 2007 and 2021. These data were acquired via 254 publicly-available reports and (to our knowledge) comprise all publicly-available EGRA data in this period that report one of the four reading skills measures analyzed in this study and that include pupils learning to read in an alphabetic writing system. Some of the EGRA surveys included in the analysis use a sampling strategy designed to be nationally-representative; this is indicated in the EGRA database made available with this article. |
| Sampling strategy | Data from existing EGRA surveys were sought from international development agencies and partner organisations. To our knowledge, our sample includes all publicly-available EGRA data in the period between 2007 and 2021 that report one of the four reading skills measures analysed in this study and that include pupils learning to read in an alphabetic writing system. |
| Data collection | This study analysed publicly-available EGRA survey data regarding foundational reading skills in low- and middle-income countries. The underlying data included reading skills measures of letter name knowledge, letter sound knowledge, nonword reading and oral reading fluency. General protocols for the construction and administration of these tests are available in the EGRA Toolkit published by RTI International (2016), while detailed descriptions of data collection protocols for each of the 230 EGRA surveys and 694 sub-surveys are available in the publicly-available reports describing those datasets. |
| Timing | The data aggregation sought publicly-available EGRA surveys undertaken between 2007 and 2021. |
| Data exclusions | Exclusions are described clearly in the manuscript. Four datapoints (around 0.1% of the dataset being analysed) were excluded because the reading assessments were not conducted to specification. |
| Non-participation | No new participants were tested so this is not relevant. |
| Randomization | There were no experimental groups in this study. |

# Ecological, evolutionary & environmental sciences study design

All studies must disclose on these points even when the disclosure is negative.

| | |
|---|---|
| Study description | *Briefly describe the study. For quantitative data include treatment factors and interactions, design structure (e.g. factorial, nested, hierarchical), nature and number of experimental units and replicates.* |
| Research sample | *Describe the research sample (e.g. a group of tagged Passer domesticus, all Stenocereus thurberi within Organ Pipe Cactus National Monument), and provide a rationale for the sample choice. When relevant, describe the organism taxa, source, sex, age range and any manipulations. State what population the sample is meant to represent when applicable. For studies involving existing datasets, describe the data and its source.* |
| Sampling strategy | *Note the sampling procedure. Describe the statistical methods that were used to predetermine sample size OR if no sample-size calculation was performed, describe how sample sizes were chosen and provide a rationale for why these sample sizes are sufficient.* |
| Data collection | *Describe the data collection procedure, including who recorded the data and how.* |
| Timing and spatial scale | *Indicate the start and stop dates of data collection, noting the frequency and periodicity of sampling and providing a rationale for these choices. If there is a gap between collection periods, state the dates for each sample cohort. Specify the spatial scale from which the data are taken* |
| Data exclusions | *If no data were excluded from the analyses, state so OR if data were excluded, describe the exclusions and the rationale behind them, indicating whether exclusion criteria were pre-established.* |
| Reproducibility | *Describe the measures taken to verify the reproducibility of experimental findings. For each experiment, note whether any attempts to repeat the experiment failed OR state that all attempts to repeat the experiment were successful.* |
| Randomization | *Describe how samples/organisms/participants were allocated into groups. If allocation was not random, describe how covariates were controlled. If this is not relevant to your study, explain why.* |
| Blinding | *Describe the extent of blinding used during data acquisition and analysis. If blinding was not possible, describe why OR explain why blinding was not relevant to your study.* |

Did the study involve field work? ☐ Yes ☐ No

# Field work, collection and transport

| | |
|---|---|
| Field conditions | *Describe the study conditions for field work, providing relevant parameters (e.g. temperature, rainfall).* |
| Location | *State the location of the sampling or experiment, providing relevant parameters (e.g. latitude and longitude, elevation, water depth).* |
| Access & import/export | *Describe the efforts you have made to access habitats and to collect and import/export your samples in a responsible manner and in compliance with local, national and international laws, noting any permits that were obtained (give the name of the issuing authority, the date of issue, and any identifying information).* |
| Disturbance | *Describe any disturbance caused by the study and how it was minimized.* |

# Reporting for specific materials, systems and methods

We require information from authors about some types of materials, experimental systems and methods used in many studies. Here, indicate whether each material, system or method listed is relevant to your study. If you are not sure if a list item applies to your research, read the appropriate section before selecting a response.

## Materials & experimental systems

| n/a | Involved in the study |
|---|---|
| ☒ | ☐ Antibodies |
| ☒ | ☐ Eukaryotic cell lines |
| ☒ | ☐ Palaeontology and archaeology |
| ☒ | ☐ Animals and other organisms |
| ☒ | ☐ Clinical data |
| ☒ | ☐ Dual use research of concern |
| ☒ | ☐ Plants |

## Methods

| n/a | Involved in the study |
|---|---|
| ☒ | ☐ ChIP-seq |
| ☒ | ☐ Flow cytometry |
| ☒ | ☐ MRI-based neuroimaging |

## Antibodies

| | |
|---|---|
| Antibodies used | *Describe all antibodies used in the study; as applicable, provide supplier name, catalog number, clone name, and lot number.* |
| Validation | *Describe the validation of each primary antibody for the species and application, noting any validation statements on the manufacturer's website, relevant citations, antibody profiles in online databases, or data provided in the manuscript.* |

## Eukaryotic cell lines

Policy information about cell lines and Sex and Gender in Research

| | |
|---|---|
| Cell line source(s) | *State the source of each cell line used and the sex of all primary cell lines and cells derived from human participants or vertebrate models.* |
| Authentication | *Describe the authentication procedures for each cell line used OR declare that none of the cell lines used were authenticated.* |
| Mycoplasma contamination | *Confirm that all cell lines tested negative for mycoplasma contamination OR describe the results of the testing for mycoplasma contamination OR declare that the cell lines were not tested for mycoplasma contamination.* |
| Commonly misidentified lines (See ICLAC register) | *Name any commonly misidentified cell lines used in the study and provide a rationale for their use.* |

## Palaeontology and Archaeology

| | |
|---|---|
| Specimen provenance | *Provide provenance information for specimens and describe permits that were obtained for the work (including the name of the issuing authority, the date of issue, and any identifying information). Permits should encompass collection and, where applicable, export.* |
| Specimen deposition | *Indicate where the specimens have been deposited to permit free access by other researchers.* |
| Dating methods | *If new dates are provided, describe how they were obtained (e.g. collection, storage, sample pretreatment and measurement), where* |

| Dating methods | *they were obtained (i.e. lab name), the calibration program and the protocol for quality assurance OR state that no new dates are provided.* |

☐ Tick this box to confirm that the raw and calibrated dates are available in the paper or in Supplementary Information.

| Ethics oversight | *Identify the organization(s) that approved or provided guidance on the study protocol, OR state that no ethical approval or guidance was required and explain why not.* |

Note that full information on the approval of the study protocol must also be provided in the manuscript.

# Animals and other research organisms

Policy information about studies involving animals; ARRIVE guidelines recommended for reporting animal research, and Sex and Gender in Research

| Laboratory animals | *For laboratory animals, report species, strain and age OR state that the study did not involve laboratory animals.* |
| Wild animals | *Provide details on animals observed in or captured in the field; report species and age where possible. Describe how animals were caught and transported and what happened to captive animals after the study (if killed, explain why and describe method; if released, say where and when) OR state that the study did not involve wild animals.* |
| Reporting on sex | *Indicate if findings apply to only one sex; describe whether sex was considered in study design, methods used for assigning sex. Provide data disaggregated for sex where this information has been collected in the source data as appropriate; provide overall numbers in this Reporting Summary. Please state if this information has not been collected.  Report sex-based analyses where performed, justify reasons for lack of sex-based analysis.* |
| Field-collected samples | *For laboratory work with field-collected samples, describe all relevant parameters such as housing, maintenance, temperature, photoperiod and end-of-experiment protocol OR state that the study did not involve samples collected from the field.* |
| Ethics oversight | *Identify the organization(s) that approved or provided guidance on the study protocol, OR state that no ethical approval or guidance was required and explain why not.* |

Note that full information on the approval of the study protocol must also be provided in the manuscript.

# Clinical data

Policy information about clinical studies
All manuscripts should comply with the ICMJE guidelines for publication of clinical research and a completed CONSORT checklist must be included with all submissions.

| Clinical trial registration | *Provide the trial registration number from ClinicalTrials.gov or an equivalent agency.* |
| Study protocol | *Note where the full trial protocol can be accessed OR if not available, explain why.* |
| Data collection | *Describe the settings and locales of data collection, noting the time periods of recruitment and data collection.* |
| Outcomes | *Describe how you pre-defined primary and secondary outcome measures and how you assessed these measures.* |

# Dual use research of concern

Policy information about dual use research of concern

## Hazards

Could the accidental, deliberate or reckless misuse of agents or technologies generated in the work, or the application of information presented in the manuscript, pose a threat to:

| No | Yes | |
|----|-----|---|
| ☒ | ☐ | Public health |
| ☒ | ☐ | National security |
| ☒ | ☐ | Crops and/or livestock |
| ☒ | ☐ | Ecosystems |
| ☒ | ☐ | Any other significant area |

## Experiments of concern

Does the work involve any of these experiments of concern:

No | Yes
☒ ☐ Demonstrate how to render a vaccine ineffective
☒ ☐ Confer resistance to therapeutically useful antibiotics or antiviral agents
☒ ☐ Enhance the virulence of a pathogen or render a nonpathogen virulent
☒ ☐ Increase transmissibility of a pathogen
☒ ☐ Alter the host range of a pathogen
☒ ☐ Enable evasion of diagnostic/detection modalities
☒ ☐ Enable the weaponization of a biological agent or toxin
☒ ☐ Any other potentially harmful combination of experiments and agents

# Plants

**Seed stocks**
*Report on the source of all seed stocks or other plant material used. If applicable, state the seed stock centre and catalogue number. If plant specimens were collected from the field, describe the collection location, date and sampling procedures.*

**Novel plant genotypes**
*Describe the methods by which all novel plant genotypes were produced. This includes those generated by transgenic approaches, gene editing, chemical/radiation-based mutagenesis and hybridization. For transgenic lines, describe the transformation method, the number of independent lines analyzed and the generation upon which experiments were performed. For gene-edited lines, describe the editor used, the endogenous sequence targeted for editing, the targeting guide RNA sequence (if applicable) and how the editor was applied.*

**Authentication**
*Describe any authentication procedures for each seed stock used or novel genotype generated. Describe any experiments used to assess the effect of a mutation and, where applicable, how potential secondary effects (e.g. second site T-DNA insertions, mosiacism, off-target gene editing) were examined.*

# ChIP-seq

## Data deposition

☐ Confirm that both raw and final processed data have been deposited in a public database such as GEO.

☐ Confirm that you have deposited or provided access to graph files (e.g. BED files) for the called peaks.

**Data access links**
*May remain private before publication.*
*For "Initial submission" or "Revised version" documents, provide reviewer access links. For your "Final submission" document, provide a link to the deposited data.*

**Files in database submission**
*Provide a list of all files available in the database submission.*

**Genome browser session**
(e.g. UCSC)
*Provide a link to an anonymized genome browser session for "Initial submission" and "Revised version" documents only, to enable peer review. Write "no longer applicable" for "Final submission" documents.*

## Methodology

**Replicates**
*Describe the experimental replicates, specifying number, type and replicate agreement.*

**Sequencing depth**
*Describe the sequencing depth for each experiment, providing the total number of reads, uniquely mapped reads, length of reads and whether they were paired- or single-end.*

**Antibodies**
*Describe the antibodies used for the ChIP-seq experiments; as applicable, provide supplier name, catalog number, clone name, and lot number.*

**Peak calling parameters**
*Specify the command line program and parameters used for read mapping and peak calling, including the ChIP, control and index files used.*

**Data quality**
*Describe the methods used to ensure data quality in full detail, including how many peaks are at FDR 5% and above 5-fold enrichment.*

**Software**
*Describe the software used to collect and analyze the ChIP-seq data. For custom code that has been deposited into a community repository, provide accession details.*

# Flow Cytometry

## Plots

Confirm that:

☐ The axis labels state the marker and fluorochrome used (e.g. CD4-FITC).

☐ The axis scales are clearly visible. Include numbers along axes only for bottom left plot of group (a 'group' is an analysis of identical markers).

☐ All plots are contour plots with outliers or pseudocolor plots.

☐ A numerical value for number of cells or percentage (with statistics) is provided.

## Methodology

| | |
|---|---|
| Sample preparation | *Describe the sample preparation, detailing the biological source of the cells and any tissue processing steps used.* |
| Instrument | *Identify the instrument used for data collection, specifying make and model number.* |
| Software | *Describe the software used to collect and analyze the flow cytometry data. For custom code that has been deposited into a community repository, provide accession details.* |
| Cell population abundance | *Describe the abundance of the relevant cell populations within post-sort fractions, providing details on the purity of the samples and how it was determined.* |
| Gating strategy | *Describe the gating strategy used for all relevant experiments, specifying the preliminary FSC/SSC gates of the starting cell population, indicating where boundaries between "positive" and "negative" staining cell populations are defined.* |

☐ Tick this box to confirm that a figure exemplifying the gating strategy is provided in the Supplementary Information.

# Magnetic resonance imaging

## Experimental design

| | |
|---|---|
| Design type | *Indicate task or resting state; event-related or block design.* |
| Design specifications | *Specify the number of blocks, trials or experimental units per session and/or subject, and specify the length of each trial or block (if trials are blocked) and interval between trials.* |
| Behavioral performance measures | *State number and/or type of variables recorded (e.g. correct button press, response time) and what statistics were used to establish that the subjects were performing the task as expected (e.g. mean, range, and/or standard deviation across subjects).* |

## Acquisition

| | |
|---|---|
| Imaging type(s) | *Specify: functional, structural, diffusion, perfusion.* |
| Field strength | *Specify in Tesla* |
| Sequence & imaging parameters | *Specify the pulse sequence type (gradient echo, spin echo, etc.), imaging type (EPI, spiral, etc.), field of view, matrix size, slice thickness, orientation and TE/TR/flip angle.* |
| Area of acquisition | *State whether a whole brain scan was used OR define the area of acquisition, describing how the region was determined.* |

Diffusion MRI      ☐ Used      ☐ Not used

## Preprocessing

| | |
|---|---|
| Preprocessing software | *Provide detail on software version and revision number and on specific parameters (model/functions, brain extraction, segmentation, smoothing kernel size, etc.).* |
| Normalization | *If data were normalized/standardized, describe the approach(es): specify linear or non-linear and define image types used for transformation OR indicate that data were not normalized and explain rationale for lack of normalization.* |
| Normalization template | *Describe the template used for normalization/transformation, specifying subject space or group standardized space (e.g. original Talairach, MNI305, ICBM152) OR indicate that the data were not normalized.* |
| Noise and artifact removal | *Describe your procedure(s) for artifact and structured noise removal, specifying motion parameters, tissue signals and physiological signals (heart rate, respiration).* |

| Volume censoring | *Define your software and/or method and criteria for volume censoring, and state the extent of such censoring.* |
|---|---|

## Statistical modeling & inference

| Model type and settings | *Specify type (mass univariate, multivariate, RSA, predictive, etc.) and describe essential details of the model at the first and second levels (e.g. fixed, random or mixed effects; drift or auto-correlation).* |
|---|---|
| Effect(s) tested | *Define precise effect in terms of the task or stimulus conditions instead of psychological concepts and indicate whether ANOVA or factorial designs were used.* |

Specify type of analysis:   ☐ Whole brain   ☐ ROI-based   ☐ Both

| Statistic type for inference<br><br>(See Eklund et al. 2016) | *Specify voxel-wise or cluster-wise and report all relevant parameters for cluster-wise methods.* |
|---|---|
| Correction | *Describe the type of correction and how it is obtained for multiple comparisons (e.g. FWE, FDR, permutation or Monte Carlo).* |

## Models & analysis

| n/a | Involved in the study |
|---|---|
| ☐ | ☐ Functional and/or effective connectivity |
| ☐ | ☐ Graph analysis |
| ☐ | ☐ Multivariate modeling or predictive analysis |

| Functional and/or effective connectivity | *Report the measures of dependence used and the model details (e.g. Pearson correlation, partial correlation, mutual information).* |
|---|---|
| Graph analysis | *Report the dependent variable and connectivity measure, specifying weighted graph or binarized graph, subject- or group-level, and the global and/or node summaries used (e.g. clustering coefficient, efficiency, etc.).* |
| Multivariate modeling and predictive analysis | *Specify independent variables, features extraction and dimension reduction, model, training and evaluation metrics.* |

