## [Peer Review File · Nature Human Behaviour]

Peer Review Information

Journal: Nature Human Behaviour

Manuscript Title: Inadequate Foundational Decoding Skills Constrain Global Literacy Goals for Pupils in Low- and Middle-Income Countries

Corresponding author name(s): Kathleen Rastle

Reviewer Comments & Decisions:

Decision Letter, initial version:

19th December 2023

Dear Professor Rastle,

Thank you once again for your manuscript, entitled "Global Literacy Goals Constrained by Inadequate Foundational Decoding Skills for Pupils in Low- and Middle-Income Countries," and for your patience during the peer review process.

Your manuscript has now been evaluated by 4 reviewers, whose comments are included at the end of this letter. Although the reviewers find your work to be of interest, they also raise some important concerns. We are interested in the possibility of publishing your study in Nature Human Behaviour, but would like to consider your response to these concerns in the form of a revised manuscript before we make a decision on publication.

To guide the scope of the revisions, the editors discuss the referee reports in detail within the team, including with the chief editor, with a view to (1) identifying key priorities that should be addressed in revision and (2) overruling referee requests that are deemed beyond the scope of the current study. We hope that you will find the prioritised set of referee points to be useful when revising your study. Please do not hesitate to get in touch if you would like to discuss these issues further.

In particular, your revision must address the following (as well as all other reviewer comments):

(1) Please provide more detailed information for the database development, including data access approval. Make sure definitions and measures are consistent and clear.

(2) Reviewer 4 raises concerns regarding the suitability of the statistical approach, as well as several analytical choices. The comments this reviewer makes should be addressed thoroughly, as inappropriate statistical analyses would fundamentally undermine the soundness of your analyses.

(3) Reviewer 3 questions the absence of analyses of data on oral language, which are key for accounting for reading difficulties and available in your dataset. We ask that your revised manuscript includes these data.

In sum, we invite you to revise your manuscript taking into account all reviewer and editor comments. We are committed to providing a fair and constructive peer-review process. Do not hesitate to contact us if there are specific requests from the reviewers that you believe are technically impossible or unlikely to yield a meaningful outcome.

We hope to receive your revised manuscript within four months. I would be grateful if you could contact us as soon as possible if you foresee difficulties with meeting this target resubmission date.

- Include a “Response to the editors and reviewers” document detailing, point-by-point, how you addressed each editor and referee comment. If no action was taken to address a point, you must provide a compelling argument. When formatting this document, please respond to each reviewer comment individually, including the full text of the reviewer comment verbatim followed by your response to the individual point. This response will be used by the editors to evaluate your revision and sent back to the reviewers along with the revised manuscript.
- Highlight all changes made to your manuscript or provide us with a version that tracks changes.

[REDACTED]

We look forward to seeing the revised manuscript and thank you for the opportunity to review your work. Please do not hesitate to contact me if you have any questions or would like to discuss these revisions further.

Sincerely,

[REDACTED]

Reviewer expertise:

Reviewer #1: language skills, children literacy

Reviewer #2: learning and schooling, LMICs research

Reviewer #3: early literacy, DIBELS

Reviewer #4: Bayesian analysis

REVIEWER COMMENTS:

Reviewer #1:

Remarks to the Author:

Thank you for the opportunity to review this interesting and I think important paper. It concerns a large-scale analysis of the extent to which reading is being effectively taught in Low-Middle Income Countries (LMICs), drawing on Early Grade Reading Assessment (EGRA) data from 48 countries. I believe the study stands to make a value contribution to global literacy debates but would like to see some refinements/revisions to improve the quality of the arguments and clarity of the discussion.

Key results

The results appear to show a significant short-fall in reading instruction in LMICs, however in places, I feel there could be some moderation of the certainty in the language around this conclusion, given the methodological limitations of the analytic approach (many of which were of course beyond the control of the researchers).

Validity

There are some issues pertaining to validity that I have highlighted under the “Suggested Improvements” heading.

Significance

Reading instruction in LMICs is a matter of global importance – it matters for the LMICs themselves and for economic prosperity and mobility across nations.

Data and methodology

I would like to see additional information on a number of points:

1. What ethical processes / approvals guided the researchers’ access to the data?
2. Reference is made (p. 15) to the data being “acquired” but it is not clear how this happened.

3. How many and which alphabetic writing systems were included in the analysis?
4. How was the term “proficient readers” operationally defined?
5. The surveys analysed spanned the years 2007 to 2021 (a 14-year period). How confident can the authors be that no policy changes were made with respect to reading instruction in any of the 62 LMICs in that time-period? This point requires some consideration in the Discussion.
6. P. 4: “Unlike DIBELS, EGRA surveys are not overseen by a centralized assessment agency but are instead produced by education officials or stakeholders in the countries where they are used”. Although implied here, this requires some specific comment in the Discussion with respect to the fact that EGRA surveys are likely to lack the psychometric robustness of tools such as DIBELS, and this has implications for the conclusions drawn.

Analytical approach

I do not feel qualified to critique the analytical approach as I have not carried out work of this scale myself. I could not see glaring issues with the analysis but recommend that a reviewer who is experienced in managing “big data” also reviews the MS.

Suggested improvements

ABSTRACT

1. Reference is made to both “learning to read” and “reading for comprehension” in the first two sentences. This creates an implication that these are distinct processes, which is not the case. It would be helpful to tighten the use of language around this.
2. It is stated that “Policymakers will reach LMICs literacy goals...”. I don’t think it’s policy makers who “reach” goals; rather, they oversee the strategic frameworks that promote/inhibit the work of others (in this case teachers) in doing so.
3. There is no mention in the Abstract as to the language(s) of instruction and/or testing in the LMICs studied. I inferred that testing is carried out in the home language in the EGRA process, but it would be helpful to state this in both the Abstract and the Introduction and Method.
4. It would be helpful to know which alphabetic writing systems were studied so that readers have some awareness of the degrees of orthographic transparency that are reflected in the data.

BODY OF THE PAPER

5. In the Abstract, learning to read is described as an “outcome” and in the opening line of the MS, it is described as a “milestone”. These are not synonymous terms, and “milestone” implies a biologically primary skill like walking, that emerges spontaneously, but that is not the case for reading (the authors are referred to the work of David Geary and others on this point).
6. P. 3 - References are needed at the end of the opening para.
7. P.3, para 2 - “However” is not a strong start to a para, as it is a conjunction.
8. As noted above in relation to the Abstract, the reader needs to know the language(s) of instruction. It would also be helpful to be given examples of LMICs in question.
9. The modifier “very” is used in a number of places in the MS but does not add value meaning-wise. I suggest removing it in all cases.
10. On p. 3, para 3 – the factors listed in lines 3 and 4 are not completely separate from the science of reading. Teachers who are inadequately prepared by their initial teacher education for example (by virtue of a lack of exposure to the science of reading), might be said to contribute to a lower quality workforce. I suggest some re-working here. Please also see my note below (Point 16) about the term “science of reading”

which I would like to see removed and replaced with more specific commentary and related references. The “science of reading” is not a static, stand-alone entity.

11. P. 3, para 3 – I suggest the authors bring in the Simple View of Reading (Gough & Tunmer, 1986) in the latter part of this para as a way of providing a theoretical framework for the important points they are making here.

12. P. 3 - The statement at the beginning of para 4 (“in high income countries....”) is a major over-generalisation which is not (and cannot be) supported by evidence. It is not possible to “lump together” the UK, USA, Canada, New Zealand and Australia on a variable such as this, as practices vary widely between and in many cases within these jurisdictions.

13. P. 3, para 4 – references are needed to support the claim that decoding skills are not assessed in LMICs. This seems like a huge generalisation (across 62 countries) that would be difficult to support with appropriate citations.

14. P. 4, para 2 – there needs to be some consideration here of the impact of home languages other than English spoken in LMICs and the fact that many children are learning in bilingual environments, which could potentially be expected to slow their progress on early markers of decoding success.

15. What did the process (mentioned on p. 16) of careful scrutinizing of the data to identify deviations of practice in the administration of EGRA surveys involve? This needs to be described in enough detail for other researchers to replicate it.

16. P. 14, para 2 – the term “the science of reading” is used in a couple of places as if it is a static entity, which it is not. I would prefer to see this re-phrased in a more qualified and specific way in both cases, e.g., (1) “according to cognitive science research, learning to read for meaning....xxxxx (refs)” and (2) “....but it is disconnected from research which indicates that xxxxx (refs)”. A similar issue occurs on p. 15. Instead of “The science of reading provides a clear pathway” please refer to particular studies and their actual findings to support the point being made.

17. The acronyms RTI and OSF are used in a number of places but are not introduced in full at first mention (p. 15 in both cases I believe).

Clarity and context

Please see above – most of my comments above could sit under this heading as well.

Pamela Snow
La Trobe University
AUSTRALIA

Reviewer #2:

Remarks to the Author:

This paper aggregates and analyzes data from hundreds of early grade ready assessments, covering hundreds of thousands of pupils, contributing to our understanding of student learning levels and gains, and the gap to reach even low level benchmarks. It contributes to our understanding of student decoding skills in low- and middle-income countries and the need for greater focus on foundational reading competencies in the early grades. This level of granularity in reading competencies data is unusual, and having comparable

data from dozens of countries together in a single database makes it a unique contribution.

It contributes in other ways too, and I have made suggestions below on some areas where I believe the authors could reinforce their contribution to the literature and help the paper reach a general interest audience. One contribution is through the creation of the database itself, as a public good which can be analyzed, including for other purposes, by others. Another contribution (which the authors do not discuss but I suggest they should) is by showing the importance of using absolute measures of learning and learning gains, rather than the current reliance on relative measures (SDs). (See note below.) Below are comments that I hope will help improve the papers.

- The authors only briefly mention differences in languages, but these differences are critical when assessing words read per minute and considering phonics approaches. I would recommend a book chapter by Elizabeth Pretorius, “Getting it right from the start: early reading instruction in African languages” to the authors. The authors should discuss the implications of language differences for the interpretation of their results. For instance, in languages with conjunctive orthographies, reading four words can take as much effort as reading eight words in English. I do not believe these differences detract from the importance of the paper’s findings, but I do believe they have a place in the interpretation, particularly against the English language benchmarks, and in the discussion. Could the authors classify the languages included in their analysis, for instance, and simulate alternative benchmarks?
- The findings on pupil progression across instructional years have close ties to work on learning trajectories, which analyze learning gains as children progress from grade to grade. The authors could link their findings to this literature, as they are offering related analysis with a new data source. See, for instance, this special issue (<https://www.sciencedirect.com/journal/international-journal-of-educational-development/special-issue/1035CNWP9N3>). In particular Kaffenberger & Pritchett’s paper developing a structural model of the learning process simulates the dynamics of flattening learning profiles, which aligns with the authors’ findings that while for most competencies children gain skills from grade to grade, they still fall further behind the benchmark because while they are indeed learning, they are learning too little to keep up or catch up.
- In Figure 1, there are large differences in number of countries included and number of languages covered for each subskill plot. Two points: 1) why the large variation; did some assessments simply not cover all of the core assessment subskills? 2) Did the authors run the analysis for only those country/language combinations for which there is data for all four subskills? Are there any differences in results?
- Another contribution this paper makes to the literature is highlighting the importance and value of using absolute measures of learning and learning gains, such as cwpm, rather than over-relying on relative measures, as the current standard practice of SDs to measure gains. Table 2 makes this point clear – from Year 1 to 2 students gain on average 0.92 standard deviations which is huge as an effect size, but this only equates to 6.37 cwpm which is tiny relative to a benchmark that expects around 80 cwpm to avoid being at “substantial risk”. I think this is an angle the authors could draw greater attention to. If more studies measured impacts in absolute terms, using some of these indicators, that could draw greater attention to the specific needs in early grade reading (like decoding/phonics), and paint a more accurate picture of gains from different intervention types. The authors could also consider bringing this point into the introduction/motivation.
- In Table 2, if the prior distribution is tied to the benchmark, and performance (the posterior distribution) is underperforming the benchmark, shouldn’t all the effect sizes be negative?
- On reading comprehension, more details on how comprehension was measured would be helpful. In Figure

2, it appears that the Y axis is # of correct items on a comprehension assessment, but the number of possible items varies across assessments. I would recommend using % correct instead so the assessments are more comparable.

- Minor point, but the authors could consider mentioning the Global Proficiency Framework as contributing to the motivation in the introduction. The motivation for this analysis is not just that decoding and phonics are important, but also that parts of the global education sector are not taking it seriously and/or prioritizing other things.

Reviewer #3:

Remarks to the Author:

Thank you for the opportunity to review the manuscript entitled Global Literacy Goals Constrained by Inadequate Foundational Decoding Skills for Pupils in Low- and Middle-Income Countries. This is an important manuscript with an important message, namely that phonics instruction must happen early in schooling for children living in LMICs to develop reading ability that will allow them to thrive. Overall I found this manuscript excellent and well written. I have one major comment and a few minor points.

The main concern I have with the article is that it singularly focuses on code-related skill development and doesn't examine children's early oral language skills (i.e., syntax, vocabulary, listening comprehension). A cursory reading of the abstract and article might leave one with the mistaken assumption that systematic phonics instruction is all that is necessary for reading. While phonics instruction is an essential component indeed, I wonder why the authors did not also examine EGRA data pertaining to oral language when it seems to be in the dataset. Low oral language is a key issue in the US related to reading difficulties. If the authors do not want to include these data in this paper, I recommend expanding the discussion a bit to discuss the importance of oral language to reading comprehension (perhaps in paragraph 5 of the discussion that appears on pp.14-15), so the reader who may not be steeped in the literature is not left with the idea that phonics instruction is all that is needed.

My other concerns are minor and easily addressed. For DIBELS, I recommend providing some background on how benchmark and at risk scores were constructed to assure the reader that the use of these criteria is appropriate for the population under study. Also, I recommend adding citations to reflect the research that support the use of these criteria beyond the administration and scoring guide. (I see that more is said about this in the Discussion but I think this information should appear earlier.)

Additional brief methods information would be helpful to the reader who reads this manuscript in the order it is presented (with Method appearing after Discussion). Please emphasize that students were taking these tests in their home language and not in English earlier in the manuscript as this is central to the manuscript. In the Method, I would like more information about the psychometric properties of the EGRA. Which countries were included in the assessments and how were the assessments conducted? What was the variability in how the scores were obtained across countries? This would give the reader heightened confidence in the data presented.

Figure 1A and Supplementary Figure 2A and 2D do not have severe or substantial risk lines in Year 3. I am assuming they are not available but am not sure why.

Reference number 15 appears incomplete in the authorship.

Reviewer #4:

Remarks to the Author:

I have focused on reviewing the analysis in this manuscript. The authors used Bayesian methods, primarily implemented by the R package “BEST”, originated from the paper of reference 12. I have some comments as follows.

Please report the version of the “BEST” package used in the analysis. It seems that this package is currently not available from the CRAN (<https://cran.r-project.org/package=BEST>), showing that “Package ‘BEST’ was removed from the CRAN repository.”

Although I understand that the authors can pass the data to the package for running the analyses, it would be helpful to specify the exact hierarchical model used in the Bayesian method, so that readers could better understand the approach. For example, what are the likelihoods of the data, and how the underlying parameters are connected?

“Data are described with the t-distribution, which has fatter tails than the normal distribution and can thus better accommodate outliers.” I wonder how the degrees of freedom were determined in the assumed t distribution. Are they modeled as ν in the following prior specification? If so, I’m not sure why the authors mentioned: “ ν is the measure of normality (in frequentist approaches, this parameter is typically referred to as degrees of freedom)”? I thought that, in Bayesian models, the degrees of freedom determine the t-distributions in the same way as frequentist methods.

“[W]e used minimally informative priors defined in [12] reflecting our lack of commitment regarding effect size and direction, with $\mu \sim \text{Normal}(\text{mean}(y), 1000 \times \text{sd}(y))$, $\sigma \sim \text{Uniform}(\text{sd}(y)/1000, \text{sd}(y) \times 1000)$, and $\nu \sim \text{Exponential}(1/29) + 1$.” It seems that the authors used the data y to determine the hyperparameters of priors for μ and σ . I don’t think this is correct, as the prior distributions should not be driven by the current data; they may be from external data or experts’ opinions. As the authors tried to use non- or weakly-informative priors, they may simply specify a normal distribution with mean 0 and a huge variance for μ and a uniform distribution from 0 to a large upper bound for σ . The choice of exponential(1/29) in the prior for ν seems arbitrary; I’m not sure this can be considered a “minimally informative prior.”

I feel that the hierarchical model would be quite simple, as the task is just to estimate the t-statistic in a Bayesian way. As the authors just used “minimally informative priors,” I wonder what were the benefits of performing the Bayesian methods over the conventional frequentist t-tests?

My concern about the data is that they may not be normally distributed or t distributed. For Figure 1, it seems that the distribution could be bimodal and highly skewed. I don’t think the Bayesian model based on t distributions can solve this problem, as the t distributions are still symmetric. I would pursue more robust methods that may be less affected by potential violations of distributional assumptions (e.g., nonparametric

methods, such as rank-based tests or permutation tests).

“The estimate of the effect size was computed for each credible combination of means and standard deviations from the posterior using the following formula (see [12] for more detail): $(\mu_1 - \mu_2) / \sqrt{(\sigma_1^2 + \sigma_2^2)/2}$.” For the denominator that takes an average of the two variances, reference 12 has a note, mentioning that many researchers use sample-size-weighted average, i.e., $(n_1 - 1) \sigma_1^2 + (n_2 - 1) \sigma_2^2 / (n_1 + n_2 - 2)$. I thought that this weighting is more common, as it is used in the conventional t-test assuming unequal variances. It also seems more sensible if the groups’ sample sizes differ greatly. For example, if group 2’s sample size is very small, say 10, and its variance estimate could be subject to large uncertainties, while group 1’s sample size is huge, say 1000, and its variance estimate is much more accurate. If taking a simple average of these two variances, the poor variance from group 2 may seriously contaminate the overall variance. Not sure if this practice is common in other relevant papers besides reference 12?

In addition, is the above equation commonly referred to as Cohen’s d? It seems that Cohen’s d usually assumes a common variance for the two groups.

Author Rebuttal to Initial comments

REVIEWER 1

Thank you for the opportunity to review this interesting and I think important paper. It concerns a large-scale analysis of the extent to which reading is being effectively taught in Low-Middle Income Countries (LMICs), drawing on Early Grade Reading Assessment (EGRA) data from 48 countries. I believe the study stands to make a value contribution to global literacy debates but would like to see some refinements/revisions to improve the quality of the arguments and clarity of the discussion.

Key results

The results appear to show a significant short-fall in reading instruction in LMICs, however in places, I feel there could be some moderation of the certainty in the language around this conclusion, given the methodological limitations of the analytic approach (many of which were of course beyond the control of the researchers).

Validity

There are some issues pertaining to validity that I have highlighted under the “Suggested Improvements” heading.

Significance

Reading instruction in LMICs is a matter of global importance – it matters for the LMICs themselves and for economic prosperity and mobility across nations.

Data and methodology

I would like to see additional information on a number of points:

1. What ethical processes / approvals guided the researchers’ access to the data?

The data provided in the EGRA database were acquired from 254 publicly available reports; no permissions to access the data were needed. The reference and link to each report is available in the

EGRA database, and the 254 reports used are also archived on the OSF site for this project. Ethical review is not relevant to our analyses given that the data we used are (a) publicly available and (b) averaged across study populations (i.e., no data from individual human participants were handled). Ethical protocols for conducting the EGRAs themselves are described in the EGRA Toolkit (RTI International, 2016), and many of the reports provide additional detail. We have enhanced our description of the underlying data in the revision (pp. 14-16).

2. Reference is made (p. 15) to the data being “acquired” but it is not clear how this happened.

We have clarified in the manuscript that the data were acquired from 254 publicly available reports (p. 14), and we have also checked and updated the links to these reports in the EGRA database.

3. How many and which alphabetic writing systems were included in the analysis?

The EGRA database includes assessments in 96 alphabetic writing systems. We now provide a list of these in Supplemental Information (see Supplementary Table S1).

4. How was the term “proficient readers” operationally defined?

We recognize that the term ‘proficient’ can have different technical definitions depending on the assessment framework; for example, the specific DIBELS and PIRLS benchmarks use different descriptors, assessment tasks, and benchmarking methods. However, we are using this term in an introduction written for a general audience. We believe the term is widely understood and does not need to be qualified by a technical definition that commits us to a particular assessment framework. Thus, we haven’t made any changes in response to this point.

5. The surveys analysed spanned the years 2007 to 2021 (a 14-year period). How confident can the authors be that no policy changes were made with respect to reading instruction in any of the 62 LMICs in that time-period? This point requires some consideration in the Discussion.

The nature of our analyses (15 years, 48 countries, 96 languages) does not allow us to go into detail on the approaches of individual countries in particular years. LMICs are unlikely to have the joined up curriculum and assessment policies enjoyed in high-income countries. But even if they did, and even if there were policy changes in particular countries across the 15 years represented in the EGRA database, we’re not sure how that would influence our conclusions regarding the low level of decoding performance documented in our analyses. Thus, we haven’t addressed this point in the Discussion.

6. P. 4: “Unlike DIBELS, EGRA surveys are not overseen by a centralized assessment agency but are instead produced by education officials or stakeholders in the countries where they are used”. Although implied here, this requires some specific comment in the Discussion with respect to the fact that EGRA surveys are likely to lack the psychometric robustness of tools such as DIBELS, and this has implications for the conclusions drawn.

We have added additional detail regarding protocols for EGRA administration to provide assurance of data quality (p. 15) and comparability across languages and countries (p. 4).

Analytical approach

I do not feel qualified to critique the analytical approach as I have not carried out work of this scale myself. I could not see glaring issues with the analysis but recommend that a reviewer who is

experienced in managing “big data” also reviews the MS.

Suggested improvements

ABSTRACT

1. Reference is made to both “learning to read” and “reading for comprehension” in the first two sentences. This creates an implication that these are distinct processes, which is not the case. It would be helpful to tighten the use of language around this.

Thank you, we now use the term ‘learning to read’.

2. It is stated that “Policymakers will reach LMICs literacy goals....”. I don’t think it’s policy makers who “reach” goals; rather, they oversee the strategic frameworks that promote/inhibit the work of others (in this case teachers) in doing so.

We agree and have rephrased this sentence.

3. There is no mention in the Abstract as to the language(s) of instruction and/or testing in the LMICs studied. I inferred that testing is carried out in the home language in the EGRA process, but it would be helpful to state this in both the Abstract and the Introduction and Method.

The EGRAs assess students primarily in a language of instruction. We have now included this information in the abstract and in the Introduction (p. 4) and Methods (p. 15). Limited information is available as to whether this was also pupils’ home language (and identifying pupils’ home language can itself be a difficult question in heterogeneous language regions where multiple local or indigenous languages are spoken).

4. It would be helpful to know which alphabetic writing systems were studied so that readers have some awareness of the degrees of orthographic transparency that are reflected in the data.

There are 96 alphabetic writing systems included in the EGRA database - too many to list in the abstract. We have now included a supplementary listing the languages and writing systems included in the analysis, and the EGRA survey numbers that assessed pupils in these languages (Supplementary Table S1). There are a large number of assessments conducted in English, French, Spanish, and Portuguese (as would be expected given language of instruction policies in many LMICs), but the other languages have not been well characterised in terms of their orthographic transparency (virtually all work on reading has been on English and related European languages).

BODY OF THE PAPER

5. In the Abstract, learning to read is described as an “outcome” and in the opening line of the MS, it is described as a “milestone”. These are not synonymous terms, and “milestone” implies a biologically primary skill like walking, that emerges spontaneously, but that is not the case for reading (the authors are referred to the work of David Geary and others on this point).

Thank you, we have rephrased to avoid the term ‘milestone’.

6. P. 3 - References are needed at the end of the opening para.

We have now included a reference (p. 3)

7. P.3, para 2 - “However” is not a strong start to a para, as it is a conjunction.

This has been reworked so that ‘however’ no longer starts the paragraph (p. 3).

8. As noted above in relation to the Abstract, the reader needs to know the language(s) of instruction. It would also be helpful to be given examples of LMICs in question.

There are too many languages and countries to list in the manuscript so these have now been provided as supplementary files (Supplementary Tables S1 and S2).

9. The modifier “very” is used in a number of places in the MS but does not add value meaning-wise. I suggest removing it in all cases.

We have followed the Reviewer’s advice and removed this word in almost all cases.

10. On p. 3, para 3 – the factors listed in lines 3 and 4 are not completely separate from the science of reading. Teachers who are inadequately prepared by their initial teacher education for example (by virtue of a lack of exposure to the science of reading), might be said to contribute to a lower quality workforce. I suggest some re-working here. Please also see my note below (Point 16) about the term “science of reading” which I would like to see removed and replaced with more specific commentary and related references. The “science of reading” is not a static, stand-alone entity.

We have reworked this sentence (page 3). We disagree with the suggestion to remove the term ‘science of reading’ and have explained why in response to the Reviewer’s Point 16.

11. P. 3, para 3 – I suggest the authors bring in the Simple View of Reading (Gough & Tunmer, 1986) in the latter part of this para as a way of providing a theoretical framework for the important points they are making here.

We appreciate the Reviewer’s suggestion, but this is quite an old reference, and it is ambiguous as to what is meant by the key concept of ‘decoding’. We’ve referenced the major review of the science of reading by Castles et al. (2018) here instead. This work discusses the Simple View of Reading in detail including more recent support for the proposition and its limitations (see their Box 5). We have also added a reference to Stainthorp (2020) who discusses the Simple View of Reading in a policy context.

12. P. 3 - The statement at the beginning of para 4 (“in high income countries.....”) is a major over-generalisation which is not (and cannot be) supported by evidence. It is not possible to “lump together” the UK, USA, Canada, New Zealand and Australia on a variable such as this, as practices vary widely between and in many cases within these jurisdictions.

We have rephrased to avoid this overgeneralization (pp. 3-4).

13. P. 3, para 4 – references are needed to support the claim that decoding skills are not assessed in LMICs. This seems like a huge generalisation (across 62 countries) that would be difficult to support with appropriate citations.

The World Bank’s *Learning Poverty Update* (2022) describes the scarcity of data on literacy in LMICs; for example, it indicates that 78 countries have failed to report *any data at any assessment point* over the

past 7 years for Sustainable Development Goal indicator 4.1.1. Likewise, Kaffenberger and Pritchett (2020) remark that while many LMICs maintain sophisticated information management systems, these typically cover data on school inputs and enrolments “*and include no reference to any measure of learning*”. It is highly unlikely that these countries are failing to capture basic information about reading comprehension outcomes, but regularly gather information on decoding subskills. We have rephrased this sentence and added these references (p. 4).

14. P. 4, para 2 – there needs to be some consideration here of the impact of home languages other than English spoken in LMICs and the fact that many children are learning in bilingual environments, which could potentially be expected to slow their progress on early markers of decoding success.

We have extended our discussion of the issue of home language and bilingual environments, and the related issues around benchmarks, in the Discussion (page 12-13).

15. What did the process (mentioned on p. 16) of careful scrutinizing of the data to identify deviations of practice in the administration of EGRA surveys involve? This needs to be described in enough detail for other researchers to replicate it.

The 254 published reports containing the EGRA scores included either (i) a copy of the instrument used for the assessment and/or (ii) descriptions of the EGRA sub-tasks assessed. These were compared to the standard protocols of administering each subtask outlined in the EGRA Toolkit (RTI International, 2016) to ascertain whether there were any substantive deviations from the standard administration methods (such as conducting an assessment without a time restriction when the assessment requires a ‘per minute’ score). The OSF site for this project has a record of all deviations that we found in the administration of all subtasks in the EGRA database (file labelled “Deviations in EGRA Administration”). We have now explained this process more clearly, but we highlight that there were only four data points that were deemed sufficiently problematic to warrant their removal from the analyses reported in this manuscript (p. 15-16).

16. P. 14, para 2 – the term “the science of reading” is used in a couple of places as if it is a static entity, which it is not. I would prefer to see this re-phrased in a more qualified and specific way in both cases, e.g., (1) “according to cognitive science research, learning to read for meaning....xxxxx (refs)” and (2) “...but it is disconnected from research which indicates that xxxxx (refs)”. A similar issue occurs on p. 15. Instead of “The science of reading provides a clear pathway” please refer to particular studies and their actual findings to support the point being made.

We appreciate the Reviewer’s nuance in respect of high-income countries and have changed some of this phrasing to ‘psychological research on reading’. However, one of the main ‘take-aways’ of our article for the international development community is that there is an extensive body of scientific knowledge about how children learn to read, and that this needs to be connected to literacy policy and practice in LMICs. The term ‘science of reading’ to refer to this body of knowledge is memorable and widely used; and we believe that our paper will have the most impact if readers unfamiliar with this body of knowledge can take away a short, salient phrase to describe it rather than individual pieces of research. We refer the reader to the influential article by Castles et al. (2018) who describe this body of knowledge in detail.

17. The acronyms RTI and OSF are used in a number of places but are not introduced in full at first mention (p. 15 in both cases I believe).

We have now specified the Open Science Framework (OSF) and specified that RTI is 'RTI International' (the organisation does not spell out the acronym).

Clarity and context

Please see above – most of my comments above could sit under this heading as well.

Pamela Snow

La Trobe University AUSTRALIA

REVIEWER 2

This paper aggregates and analyzes data from hundreds of early grade ready assessments, covering hundreds of thousands of pupils, contributing to our understanding of student learning levels and gains, and the gap to reach even low level benchmarks. It contributes to our understanding of student decoding skills in low- and middle-income countries and the need for greater focus on foundational reading competencies in the early grades. This level of granularity in reading competencies data is unusual, and having comparable data from dozens of countries together in a single database makes it a unique contribution.

It contributes in other ways too, and I have made suggestions below on some areas where I believe the authors could reinforce their contribution to the literature and help the paper reach a general interest audience. One contribution is through the creation of the database itself, as a public good which can be analyzed, including for other purposes, by others. Another contribution (which the authors do not discuss but I suggest they should) is by showing the importance of using absolute measures of learning and learning gains, rather than the current reliance on relative measures (SDs). (See note below.) Below are comments that I hope will help improve the papers.

- The authors only briefly mention differences in languages, but these differences are critical when assessing words read per minute and considering phonics approaches. I would recommend a book chapter by Elizabeth Pretorius, "Getting it right from the start: early reading instruction in African languages" to the authors. The authors should discuss the implications of language differences for the interpretation of their results. For instance, in languages with conjunctive orthographies, reading four words can take as much effort as reading eight words in English. I do not believe these differences detract from the importance of the paper's findings, but I do believe they have a place in the interpretation, particularly against the English language benchmarks, and in the discussion. Could the authors classify the languages included in their analysis, for instance, and simulate alternative benchmarks?

We thank the Reviewer for pointing us to this excellent book chapter. We have enhanced our discussion of language differences in the Discussion along the lines that the Reviewer suggests (p. 12). However, with 96 alphabetic systems included in the database (most of which are not well characterised), we do not believe that it would be feasible to "classify" these in a meaningful way and measure them against alternative benchmarks (even if these were available or could be straightforwardly simulated). We hope that individual researchers will use the EGRA database to ask questions about specific languages and writing systems in the way that the Reviewer suggests.

- The findings on pupil progression across instructional years have close ties to work on learning trajectories, which analyze learning gains as children progress from grade to grade. The authors could link their findings to this literature, as they are offering related analysis with a new data source. See, for

instance, this special issue (<https://www.sciencedirect.com/journal/international-journal-of-educational-development/special-issue/1035CNWP9N3>). In particular Kaffenberger & Pritchett's paper developing a structural model of the learning process simulates the dynamics of flattening learning profiles, which aligns with the authors' findings that while for most competencies children gain skills from grade to grade, they still fall further behind the benchmark because while they are indeed learning, they are learning too little to keep up or catch up.

Thank you, this work really resonates with our findings so we have added a paragraph describing it in the Discussion (pp. 11-12).

- In Figure 1, there are large differences in number of countries included and number of languages covered for each subskill plot. Two points: 1) why the large variation; did some assessments simply not cover all of the core assessment subskills? 2) Did the authors run the analysis for only those country/language combinations for which there is data for all four subskills? Are there any differences in results?

The Reviewer is correct that the different EGRA surveys often did not cover all of the subskills of interest. We lose a lot of data by requiring that the EGRA surveys use all four tasks (this brings us down to 15 countries, 12 languages, and 100 subskill scores per task). However, to confirm that the data in this subset of EGRAs follow the pattern reported for the entire set of surveys, we have carefully inspected the subskill scores in these EGRAs. These data are plotted below, and it is evident that the pattern remains the same. We have added text to this effect (p. 7), and the data are available in Supplemental Information (Figure S3 and Table S5).

Figure S3. Subskill scores by task and instructional year only for those EGRA surveys that included all four tasks. For each task, these EGRA surveys represent 100 subskill scores from 15 countries and 12 languages.

- Another contribution this paper makes to the literature is highlighting the importance and value of using absolute measures of learning and learning gains, such as cwpm, rather than over-relying on relative measures, as the current standard practice of SDs to measure gains. Table 2 makes this point clear – from Year 1 to 2 students gain on average 0.92 standard deviations which is huge as an effect size, but this only equates to 6.37 cwpm which is tiny relative to a benchmark that expects around 80 cwpm to avoid being at “substantial risk”. I think this is an angle the authors could draw greater attention to. If more studies measured impacts in absolute terms, using some of these indicators, that could draw greater attention to the specific needs in early grade reading (like decoding/phonics), and paint a more accurate picture of gains from different intervention types. The authors could also consider bringing this point into the introduction/motivation.

This is a very important point, and we are grateful to the Reviewer for raising it. As this was not part of

the *a priori* motivation for this work, we decided that the best place to highlight it is in the Discussion (pp. 11-12).

- In Table 2, if the prior distribution is tied to the benchmark, and performance (the posterior distribution) is underperforming the benchmark, shouldn't all the effect sizes be negative?

We assume that the Reviewer is referring to Table 3 as Table 2 reports effect sizes for the comparison of performance across the instructional years. We thank the Reviewer for spotting the error in these values which has now been corrected.

- On reading comprehension, more details on how comprehension was measured would be helpful. In Figure 2, it appears that the Y axis is # of correct items on a comprehension assessment, but the number of possible items varies across assessments. I would recommend using % correct instead so the assessments are more comparable.

We are grateful to the Reviewer for bringing this up. Unfortunately, ambiguities in the administration of some EGRAs (e.g., aborting the test earlier than recommended in the manual) make it difficult to derive appropriate percent correct estimates for all data points. However, because only 8% of EGRAs that assessed reading comprehension used a number of possible items that did not fall between 4 and 6 (reported in Footnote 2 of the original manuscript), we have decided to remove data from these assessments from the analysis. The removal of these extreme cases has resulted in a slight increase in all four correlation coefficients (see updated Figure 2 on page 10), but the overall pattern of results has not changed, and the assessments are now more comparable. We report this information on page 10, and we have also added more detail on how reading comprehension was assessed (page 9).

- Minor point, but the authors could consider mentioning the Global Proficiency Framework as contributing to the motivation in the introduction. The motivation for this analysis is not just that decoding and phonics are important, but also that parts of the global education sector are not taking it seriously and/or prioritizing other things.

Thank you, we have revised following the Reviewer's valuable insight and now mention this on p. 4.

REVIEWER 3

Remarks to the Author:

Thank you for the opportunity to review the manuscript entitled Global Literacy Goals Constrained by Inadequate Foundational Decoding Skills for Pupils in Low- and Middle-Income Countries. This is an important manuscript with an important message, namely that phonics instruction must happen early in schooling for children living in LMICs to develop reading ability that will allow them to thrive. Overall I found this manuscript excellent and well written. I have one major comment and a few minor points.

The main concern I have with the article is that it singularly focuses on code-related skill development and doesn't examine children's early oral language skills (i.e., syntax, vocabulary, listening comprehension). A cursory reading of the abstract and article might leave one with the mistaken assumption that systematic phonics instruction is all that is necessary for reading. While phonics instruction is an essential component indeed, I wonder why the authors did not also examine EGRA data pertaining to oral language when it seems to be in the dataset. Low oral language is a key issue in the US

related to reading difficulties. If the authors do not want to include these data in this paper, I recommend expanding the discussion a bit to discuss the importance of oral language to reading comprehension (perhaps in paragraph 5 of the discussion that appears on pp.14-15), so the reader who may not be steeped in the literature is not left with the idea that phonics instruction is all that is needed.

Thank you, we agree that oral language is an important foundation of learning to read. Although we do have listening comprehension data in the EGRA database, there is no equivalent DIBELS task and thus we lack a benchmark to interpret these data against. However, we have undertaken two additional analyses showing (a) that the relationship between listening comprehension and reading comprehension is significant but far weaker ($r < .25$) than the relationship between the four decoding tasks and reading comprehension (all r 's $> .70$); and (b) that the relationship between listening comprehension and the four decoding tasks is weak (all r 's $< .20$). Thus, while acknowledging the importance of oral language on reading acquisition, we do not believe that weaknesses in oral language are the primary driver of the poor reading outcomes that we report. We discuss these issues on page 13, and the figures below are available in Supplemental Information (Figures S4 and S5).

Figure S4. Relationship between listening comprehension and reading comprehension.

Note. Scores were included only where we also have a measure from at least one of the decoding tasks under investigation (thus ensuring that the samples reflected in this graph overlap with those reflected in Figure 1 of the manuscript).

Figure S5. Relationship between listening comprehension and the four decoding tasks.

My other concerns are minor and easily addressed. For DIBELS, I recommend providing some background on how benchmark and at risk scores were constructed to assure the reader that the use of these criteria is appropriate for the population under study. Also, I recommend adding citations to reflect the research that support the use of these criteria beyond the administration and scoring guide. (I see that more is said about this in the Discussion but I think this information should appear earlier.)

We have now included information about how DIBELS benchmarks are calculated (p. 16) and have made reference to the fact that a majority of US states now require decoding assessments to screen for reading disabilities (including 17 states that specifically mention use of DIBELS instruments; p. 4).

Additional brief methods information would be helpful to the reader who reads this manuscript in the order it is presented (with Method appearing after Discussion). Please emphasize that students were taking these tests in their home language and not in English earlier in the manuscript as this is central to the manuscript. In the Method, I would like more information about the psychometric properties of the EGRA. Which countries were included in the assessments and how were the assessments conducted?

What was the variability in how the scores were obtained across countries? This would give the reader heightened confidence in the data presented.

We have now included information on the language of testing in the abstract and in the Introduction (p. 4). We have also included additional detail on assessment protocols used to ensure comparability across countries (pp. 4,15), and have now provided in Supplemental Information a list of countries and languages included in the database (Tables S1 and S2).

Figure 1A and Supplementary Figure 2A and 2D do not have severe or substantial risk lines in Year 3. I am assuming they are not available but am not sure why.

Benchmarks are available only where a DIBELS task is regularly given in that instructional year (for example, the DIBELS letter name identification task is not given in the third instructional year because virtually all pupils would be expected to know their letters at that point). We have updated the text (pp. 5-6) to clarify this.

Reference number 15 appears incomplete in the authorship. Thank you, this has been fixed.

REVIEWER 4

Remarks to the Author:

I have focused on reviewing the analysis in this manuscript. The authors used Bayesian methods, primarily implemented by the R package “BEST”, originated from the paper of reference 12. I have some comments as follows.

Please report the version of the “BEST” package used in the analysis. It seems that this package is currently not available from the CRAN (<https://cran.r-project.org/package=BEST>), showing that “Package ‘BEST’ was removed from the CRAN repository.”

Thank you, we have reported the version of the package that we used (version 0.5.4, see page 16). We note that this and all older versions are still available for download and are fully functional.

Although I understand that the authors can pass the data to the package for running the analyses, it would be helpful to specify the exact hierarchical model used in the Bayesian method, so that readers could better understand the approach. For example, what are the likelihoods of the data, and how the underlying parameters are connected?

Thank you, we now report this information on pages 17-18 in the Methods section.

“Data are described with the t-distribution, which has fatter tails than the normal distribution and can thus better accommodate outliers.” I wonder how the degrees of freedom were determined in the assumed t distribution. Are they modeled as ν in the following prior specification? If so, I’m not sure why the authors mentioned: “ ν is the measure of normality (in frequentist approaches, this parameter is typically referred to as degrees of freedom)”? I thought that, in Bayesian models, the degrees of freedom determine the t-distributions in the same way as frequentist methods.

We agree with the Reviewer that, in the context of sampling distributions, the parameter ν that determines the shape of the probability density function of the t-distribution is traditionally referred to as the degrees of freedom. Because, in our analysis approach, we did not use the t-distribution in that context, we thought that this more traditional nomenclature could be potentially misleading and, therefore, decided to refer to ν by its effect on the distribution's shape (i.e., as the normality parameter). Nevertheless, to make the analysis description more accessible to those readers who may not have much experience with this use of the t-distribution, in the original version of the manuscript, we had added that the ν parameter is also known as the degrees of freedom. The fact that the Reviewer had trouble understanding this made us realise that, contrary to our intentions, our original description lacked clarity, and we have therefore reworked this passage (p. 17).

“[W]e used minimally informative priors defined in [12] reflecting our lack of commitment regarding effect size and direction, with $\mu \sim \text{Normal}(\text{mean}(y), 1000 \times \text{sd}(y))$, $\sigma \sim \text{Uniform}(\text{sd}(y)/1000, \text{sd}(y) \times 1000)$, and $\nu \sim \text{Exponential}(1/29) + 1$.” It seems that the authors used the data y to determine the hyperparameters of priors for μ and σ . I don't think this is correct, as the prior distributions should not be driven by the current data; they may be from external data or experts' opinions. As the authors tried to use non- or weakly-informative priors, they may simply specify a normal distribution with mean 0 and a huge variance for μ and a uniform distribution from 0 to a large upper bound for σ . The choice of $\text{exponential}(1/29)$ in the prior for ν seems arbitrary; I'm not sure this can be considered a “minimally informative prior.”

We thank the Reviewer for their careful reading of the Methods section, as the choice of priors is of crucial importance. We also agree with the Reviewer that the data should not be used for prior specification but that the choice of priors should be based on expert knowledge. However, we do not think that this criticism applies to our work: our decisions regarding the choice of priors were not arbitrary, and we followed practices attested in the expert literature throughout this process (Kruschke, 2013; 2015). We summarise our reasoning below, and we have also added additional justification for our approach on page 16.

In our analysis, the prior on the mean parameters μ_1 and μ_2 is assumed to be a broad normal distribution. As shown in the formula the Reviewer cites, in BEST, the mean of the prior on μ_1 and μ_2 is set to the mean of the pooled data, while the standard deviations of the prior on μ_1 and μ_2 are set to 1,000 times the standard deviation of the pooled data. This setting allowed us to ensure that the prior was scaled appropriately relative to the scale of the data while being so broad and vague that its influence on the estimates of the parameters was minimal, and even a modest amount of data would have overwhelmed the prior assumptions when the parameter estimation was done. Similarly, the prior on the standard deviation parameter was also assumed to be non-committal, expressed as a uniform distribution ranging from a low value (one thousandth of the standard deviation of the pooled data) to a high value (one thousand times the standard deviation of the pooled data).

Finally, the prior on the ν parameter was not chosen arbitrarily: following Kruschke (2013, 2015), the ν parameter has a prior that is exponentially distributed such that prior credibility is spread fairly evenly over nearly normal and heavy tailed data:

$$p(\nu|\lambda) = (1/\lambda) \exp[-(\nu - 1)/\lambda], \text{ where } \nu \geq 1 \text{ and } \lambda = 29$$

Here, λ was set to 29 because it balances nearly normal distributions ($\nu > 30$) with heavy tailed distributions ($\nu < 30$). Kruschke (2013) argues that this prior performs better than many other priors

(e.g., various uniform distributions, shifted gamma distributions, or shifted and folded t-distributions), and we see no theoretical reason to change this setting.

I feel that the hierarchical model would be quite simple, as the task is just to estimate the t-statistic in a Bayesian way. As the authors just used “minimally informative priors,” I wonder what were the benefits of performing the Bayesian methods over the conventional frequentist t-tests?

We believe that the choice of framework (and, consequently, one’s approach to interpreting probability and quantifying belief) should be independent of model complexity or analytical aims. In our view, the Bayesian approach to statistical inference is preferable to the frequentist approach as it is more conservative and more flexible than standard NHST, and better reflects scientific reasoning (e.g., it allows us to answer the research question itself rather than estimating the probability with which the data can be observed under the null hypothesis). Bayesian methods also allow us to quantify the uncertainty regarding the effect of interest instead of relying on point estimates to make binary decisions about the presence or absence of an effect (and we discuss this point in the paper).

Regarding the model structure, the Reviewer is correct in saying that the model we used was rather simple (now reported on page 17 of the revised manuscript). This model choice was determined by the features of the dataset, where each data point represents an *average* score for pupils included in an EGRA sub-survey for each individual subskill assessment. In many cases, we do not have enough information about the sampling method used in a given EGRA sub-survey (e.g., whether the aggregated score represents data from pupils from one or several schools, etc) to make an informed choice as to which hierarchical structure would be appropriate. This is further complicated by the fact that the data are observational rather than experimental, and there is uncertainty as to which variables should be expected to cause the observed differences. We now mention these issues on page 18 of the revised manuscript.

My concern about the data is that they may not be normally distributed or t distributed. For Figure 1, it seems that the distribution could be bimodal and highly skewed. I don’t think the Bayesian model based on t distributions can solve this problem, as the t distributions are still symmetric. I would pursue more robust methods that may be less affected by potential violations of distributional assumptions (e.g., nonparametric methods, such as rank-based tests or permutation tests).

We thank the Reviewer for expressing their concern, which we take very seriously. We have carefully re-considered the appropriateness of our analytical choices, and we believe them to be sound. We summarise our justification below; however, if our arguments do not alleviate the Reviewer’s or the Editor’s concerns, we would be happy to implement further changes.

Our (*a priori*) choice of the t-distribution to describe the data was motivated by our subject matter knowledge: we expected the aggregated scores to follow the t-distribution because we had no theoretical reason to expect the true data generating process to be bimodal or skewed. Once the models were fit, we conducted a series of posterior predictive checks that confirmed that the model fit was satisfactory in all cases. One potential issue with the t-distribution could be that it is parameterised using the mean, whereas, when the data are skewed, it may be preferable to use the median. However, in the present case, the group means are greater than the medians (and note that these differences are rather small), meaning that our analysis approach is conservative in terms of the point we are making (i.e., that there is improvement across the instructional years but that the vast majority of the scores are still below the substantial risk benchmark). Thus, our models are theoretically motivated, capture the

data adequately, and provide estimates that err on the side of caution, meaning that they are more likely to be valid under a wide range of conditions.

“The estimate of the effect size was computed for each credible combination of means and standard deviations from the posterior using the following formula (see [12] for more detail): $(\mu_1 - \mu_2) / \sqrt{(\sigma_1^2 + \sigma_2^2) / 2}$.” For the denominator that takes an average of the two variances, reference 12 has a note, mentioning that many researchers use sample-size-weighted average, i.e., $(n_1 - 1) \sigma_1^2 + (n_2 - 1) \sigma_2^2 / (n_1 + n_2 - 2)$. I thought that this weighting is more common, as it is used in the conventional t-test assuming unequal variances. It also seems more sensible if the groups’ sample sizes differ greatly. For example, if group 2’s sample size is very small, say 10, and its variance estimate could be subject to large uncertainties, while group 1’s sample size is huge, say 1000, and its variance estimate is much more accurate. If taking a simple average of these two variances, the poor variance from group 2 may seriously contaminate the overall variance. Not sure if this practice is common in other relevant papers besides reference 12?

In addition, is the above equation commonly referred to as Cohen’s d? It seems that Cohen’s d usually assumes a common variance for the two groups.

We thank the Reviewer for this insight. Cohen (1988) offers several options for calculating the pooled standard deviation, depending on whether the equality of variance can or cannot be assumed. Originally, we followed the approach developed by Kruschke (e.g., Kruschke, 2013; 2015) to compute the effect size. Under this approach, the effect size is treated as a re-description of the posterior distribution and, because a posterior distribution can in theory be generated by an almost infinite number of datasets, the sample size is not used in the effect size calculations. However, following the Reviewer’s comment, we have reconsidered our decision, and agree that sample-size weighted averaging is a more sensible approach, while it is also more commonly used. We have therefore recomputed the effect sizes in the analysis examining pupil progress across the instructional years; as is clear from Table 2, this change has not had a substantive impact on the estimates. The description of the effect size calculation has been modified accordingly (see page 18).

References

Castles, A., Rastle, K., & Nation, K. (2018). Ending the Reading Wars: Reading acquisition from novice to expert. *Psychological Science in the Public Interest*, 19, 5–51. <https://doi.org/10.1177/1529100618772271>

Cohen, J. (1988). *Statistical Power Analysis for the Behavioral Sciences* (2nd ed.). Hillsdale, NJ: Erlbaum.

Kaffenberger, M. & Pritchett, L. (2020). Aiming higher: Learning profiles and gender equality in 10 low- and middle-income countries. *International Journal of Educational Development*, 79, 102272. <https://doi.org/10.1016/j.ijedudev.2020.102272>

Kruschke, J.K. (2013). Bayesian estimation supersedes the t Test. *Journal of Experimental Psychology: General*, 142, 573-603. <https://doi.org/10.1037/a0029146>

Kruschke, J.K. (2015). *Doing Bayesian Data Analysis: A Tutorial with R, JAGS, and Stan* (2nd ed). Academic Press.

RTI International (2016). *Early Grade Reading Assessment (EGRA) Toolkit, Second Edition*. Washington, DC: United States Agency for International Development.

World Bank (2022). The state of global learning poverty: 2022 update. Retrieved from <https://www.worldbank.org/en/topic/education/publication/state-of-global-learning-poverty>

Decision Letter, first revision:

27th May 2024

Dear Professor Rastle,

Thank you once again for your revised manuscript, entitled "Global Literacy Goals Constrained by Inadequate Foundational Decoding Skills for Pupils in Low- and Middle-Income Countries". Please accept my sincere apologies for the delay in contacting you with a decision on your manuscript and thank you for your patience during the review process

Your manuscript has now been evaluated by the same reviewers who evaluated your original manuscript. All reviewer feedback is included at the end of this letter. Although the reviewers found your manuscript to have improved during revision, Reviewer #4 also raises some important outstanding concerns. We remain interested in the possibility of publishing your study in Nature Human Behaviour, but would like to consider your response to these outstanding concerns in the form of a revised manuscript before we make a decision on publication.

In particular, we would request you to provide the justification for the use of Bayesian analyses, as well as sensitivity analyses that demonstrate that results are not biased.

In sum, we invite you to revise your manuscript taking into account all reviewer and editor comments. We are committed to providing a fair and constructive peer-review process. Do not hesitate to contact us if there are specific requests from the reviewers that you believe are technically impossible or unlikely to yield a meaningful outcome.

We hope to receive your revised manuscript within 4-8 weeks. I would be grateful if you could contact us as soon as possible if you foresee difficulties with meeting this target resubmission date.

- Include a “Response to the editors and reviewers” document detailing, point-by-point, how you addressed each editor and referee comment. If no action was taken to address a point, you must provide a compelling argument. This response will be used by the editors and reviewers to evaluate your revision.
- Highlight all changes made to your manuscript or provide us with a version that tracks changes.

[REDACTED]

We look forward to seeing the revised manuscript and thank you for the opportunity to review your work. Please do not hesitate to contact me if you have any questions or would like to discuss these revisions further.

Sincerely,

[REDACTED]

Reviewer expertise:

Reviewer #1: language skills, children literacy

Reviewer #2: learning and schooling, LMICs research

Reviewer #3: early literacy, DIBELS

Reviewer #4: Bayesian analysis

REVIEWER COMMENTS:

Reviewer #1:

None

Reviewer #2:

None

Reviewer #3:

Remarks to the Author:

Thank you for the opportunity to review the revised manuscript entitled Global Literacy Goals Constrained by Inadequate Foundational Decoding Skills for Pupils in Low- and Middle-Income Countries. This is an important manuscript with a clear message, namely that phonics instruction must happen early in schooling for children living in LMICs to develop reading ability that will allow them to thrive. This study is well conceived, and the manuscript is well written. In the prior round of review, I expressed concern about the lack of attention to oral language skills. The authors have adequately responded to my concerns and include additional analyses regarding the relation between listening comprehension and reading comprehension. The authors have also addressed my other more minor recommendations. I commend the authorship team for undertaking this needed work.

Reviewer #4:

Remarks to the Author:

I thank the reviewers for carefully responding to my previous comments. Many of them have been addressed in the revised manuscript. Nevertheless, I'm still not convinced by the authors regarding why they used t-distributions to model the data, while the data appears to be skewed. The authors' responses seem to indicate that this assumption is their prior belief, but I think the authors confused the model specification with the prior specification. In Bayesian analyses, the prior distributions are assigned to parameters, while we should still respect the likelihood assumed for the data. If the data's likelihood is incorrectly specified, the results could be seriously biased. At least, I think the authors should conduct sensitivity analyses with different distributional assumptions, including skewed ones.

In addition, as the authors acknowledged, the model is very simple, and the prior are weakly informative. In this case, I don't think Bayesian analyses would provide much more benefits than frequentist analyses. Have the authors ever tried to use frequentist methods to fit the data? What would be the results? In fact, in Bayesian theories, with non-informative priors, the posterior mode estimates would be the same as the maximum likelihood estimates of frequentist analyses. I think the authors should add more motivation to justify the use of Bayesian analyses in their study.

Author Rebuttal, first revision:

Reviewer #4:

Remarks to the Author:

I thank the reviewers for carefully responding to my previous comments. Many of them have been addressed in the revised manuscript. Nevertheless, I'm still not convinced by the authors regarding why they used t-distributions to model the data, while the data appears to be skewed. The authors' responses seem to indicate that this assumption is their prior belief, but I think the authors confused the model specification with the prior specification. In Bayesian analyses, the prior distributions are assigned to parameters, while we should still respect the likelihood assumed for the data. If the data's likelihood is incorrectly specified, the results could be seriously biased. At least, I think the authors should conduct sensitivity analyses with different distributional assumptions, including skewed ones.

Please see our response to the Editor.

In addition, as the authors acknowledged, the model is very simple, and the prior are weakly informative. In this case, I don't think Bayesian analyses would provide much more benefits than frequentist analyses. Have the authors ever tried to use frequentist methods to fit the data? What would be the results? In fact, in Bayesian theories, with non-informative priors, the posterior mode estimates would be the same as the maximum likelihood estimates of frequentist analyses. I think the authors should add more motivation to justify the use of Bayesian analyses in their study.

Please see our response to the Editor.

References

Byrne, B. M. (2010). *Structural equation modeling with AMOS: Basic concepts, applications, and programming*. New York: Routledge.

Curran, P. J., West, S. G., & Finch, J. F. (1996). The robustness of test statistics to nonnormality and specification error in confirmatory factor analysis. *Psychological Methods*, 1(1), 16-29. <https://doi.org/10.1037/1082-989X.1.1.16>

Gelman, A., Carlin, J. B., Stern, H. S., Dunson, D. B., Vehtari, A., & Rubin, D. B. (2014). *Bayesian data analysis* (Third). Boca Raton, FL: Chapman; Hall/CRC.

Gelman, A., & Hill, J. (2007). *Data analysis using regression and multilevel/hierarchical models*. Cambridge: Cambridge University Press.

George, D. & Mallery, M. (2010). *SPSS for Windows Step by Step: A Simple Guide and Reference*, 17.0 update (10a ed.) Boston: Pearson.

Hair, J., Black, W. C., Babin, B. J. & Anderson, R. E. (2010). *Multivariate data analysis* (7th ed.). Upper Saddle River, New Jersey: Pearson Educational International.

Hair, J. F., Hult, G. T. M., Ringle, C. M., & Sarstedt, M. (2022). *A Primer on Partial Least Squares Structural Equation Modeling (PLS-SEM)* (3 ed.). Thousand Oaks, CA: Sage.

Hansen, B. E. (1994). Autoregressive conditional density estimation. *International Economic Review*, 35 (3), 705–730. <https://doi.org/10.2307/2527081>

Hansen, C., McDonald, J., & Newey, W. (2010). Instrumental variables estimation with flexible distributions. *Journal of Business and Economic Statistics*, 28, 13–25. <https://doi.org/10.1198/jbes.2009.06161>

Morey, R. D., Romeijn, J.-W., & Rouder, J. N. (2016). The philosophy of Bayes factors and the quantification of statistical evidence. *Journal of Mathematical Psychology*, 72, 6–18. <https://doi.org/10.1016/j.jmp.2015.11.001>

Nicenboim, B., & Vasishth, S. (2016). Statistical methods for linguistic research: Foundational Ideas -- Part II. *Language and Linguistics Compass*, 10, 591-613. <https://doi.org/10.1111/lnc3.12207>

Theodossiou, P. (1998). Financial data and the Skewed Generalized T distribution. *Management Science*, 44 (12-part-1), 1650–1661. <https://doi.org/10.1287/mnsc.44.12.1650>

Vasishth, S. (2023). Some right ways to analyze (psycho)linguistic data. *Annual Review of Linguistics*, 9, 273–291. <https://doi.org/10.1146/annurev-linguistics-031220-010345>

Vasishth, S., & Gelman, A. (2021). How to embrace variation and accept uncertainty in linguistic and psycholinguistic data analysis. *Linguistics*, 59, 1311-1342. <https://doi.org/10.1515/ling-2019-0051>

Wagenmakers, E. J., Marsman, M., Jamil, T. *et al.* (2018). Bayesian inference for psychology. Part I: Theoretical advantages and practical ramifications. *Psychonomic Bulletin & Review*, 25, 35–57. <https://doi.org/10.3758/s13423-017-1343-3>

Wagenmakers, E.-J., Morey, R. D., & Lee, M. D. (2016). Bayesian benefits for the pragmatic researcher. *Current Directions in Psychological Science*, 25(3), 169–176. <https://doi.org/10.1177/096372141664>

Decision Letter, second revision:

5th August 2024

Dear Dr. Rastle,

Thank you for your patience as we've prepared the guidelines for final submission of your Nature Human Behaviour manuscript, "Global Literacy Goals Constrained by Inadequate Foundational Decoding

Skills for Pupils in Low- and Middle-Income Countries" (NATHUMBEHAV-23082773B). Please carefully follow the step-by-step instructions provided in the attached file, and add a response in each row of the table to indicate the changes that you have made. Please also address the additional marked-up edits we have proposed within the reporting summary. Ensuring that each point is addressed will help to ensure that your revised manuscript can be swiftly handed over to our production team.

We would hope to receive your revised paper, with all of the requested files and forms within two-three weeks. Please get in contact with us if you anticipate delays.

Nature Human Behaviour offers a Transparent Peer Review option for new original research manuscripts submitted after December 1st, 2019. As part of this initiative, we encourage our authors to support increased transparency into the peer review process by agreeing to have the reviewer comments, author rebuttal letters, and editorial decision letters published as a Supplementary item. When you submit your final files please clearly state in your cover letter whether or not you would like to participate in this initiative. Please note that failure to state your preference will result in delays in accepting your manuscript for publication.

In recognition of the time and expertise our reviewers provide to Nature Human Behaviour's editorial process, we would like to formally acknowledge their contribution to the external peer review of your manuscript entitled "Global Literacy Goals Constrained by Inadequate Foundational Decoding Skills for Pupils in Low- and Middle-Income Countries". For those reviewers who give their assent, we will be publishing their names alongside the published article.

Cover suggestions

We welcome submissions of artwork for consideration for our cover. For more information, please see our guide for cover artwork.

ORCID

Non-corresponding authors do not have to link their ORCID but are encouraged to do so. Please note that it will not be possible to add/modify ORCIDs at proof. Thus, please let your co-authors know that if they wish to have their ORCID added to the paper they must follow the procedure described in the following link prior to acceptance: <https://www.springernature.com/gp/researchers/orcid/orcid-for-nature-research>

Nature Human Behaviour has now transitioned to a unified Rights Collection system which will allow our Author Services team to quickly and easily collect the rights and permissions required to publish your work. Approximately 10 days after your paper is formally accepted, you will receive an email in providing you with a link to complete the grant of rights. If your paper is eligible for Open Access, our Author Services team will also be in touch regarding any additional information that may be required to arrange payment for your article.

Please note that *Nature Human Behaviour* is a Transformative Journal (TJ). Authors may publish their research with us through the traditional subscription access route or make their paper immediately open access through payment of an article-processing charge (APC). Authors will not be required to make a final decision about access to their article until it has been accepted. Find out more about Transformative Journals

Please use the following link for uploading these materials:
[REDACTED]

Best regards,

[REDACTED]

Reviewer #4:

None

Final Decision Letter: